

# Assessment of co-benefits of black carbon emission reduction measures in Southeast Asia: Part 1 emission inventory and simulation for the base year 2007

Didin Agustian Permadi[1], Nguyen Thi Kim Oanh[1*] and Robert Vautard[2]

[1] *Environmental Engineering and Management; School of Environment, Resources and Development; Asian Institute of Technology; Klong Luang, Pathumthani 12120, Thailand.*

[2] *Laboratoire des Sciences du Climate de l'Environment (LSCE), Institut Pierre Simon Laplace (IPSL), Gif Sur Yvette, France.*

*Correspondence to*: Nguyen Thi Kim Oanh (kimoanh@ait.ac.th)

**Abstract.** This research assessed the potential co-benefits associated with selected black carbon (BC) emission reduction measures on mitigation of air pollution and climate forcing in Southeast Asia (SEA). This paper presents Part 1 of the research with details on the emission inventory (EI) results and the WRF/CHIMERE model performance evaluation. The SEA regional emissions for 2007 were updated with our EI results for Indonesia, Thailand and Cambodia and used for the model input. WRF/CHIMERE simulated $PM_{10}$, $PM_{2.5}$ and BC over the SEA domain (0.25º x 0.25º) of the year 2007 and the results were evaluated against the available monitoring data in the domain. WRF hourly simulation results were evaluated using the observed data at 8 international airport stations in 5 SEA countries and showed a satisfactory performance. WRF/CHIMERE results for $PM_{10}$ and $PM_{2.5}$ showed strong seasonal influence of biomass open burning while BC distribution showed the influence of urban activities in big SEA cities. Daily average $PM_{10}$ constructed from the hourly concentrations were obtained from the automatic monitoring stations in three SEA large cities, i.e. Bangkok, Kuala Lumpur and Surabaya for model evaluation. The daily observed $PM_{2.5}$ and BC concentrations obtained from the Improving Air Quality in the Asian Developing Countries (AIRPET) project for 4 cities (i.e. Bangkok, Hanoi, Bandung, and Manila) were also used for model evaluation. In addition, hourly BC concentrations were taken from the measurement results of the Asian Pacific Network (APN) project at a sub-urban site in Bangkok. The modeled $PM_{10}$ and BC satisfactorily met all suggested statistical criteria for PM evaluation. The modeled $PM_{2.5}/PM_{10}$ ratios estimated for four AIRPET sites ranged between 0.47 - 0.59, lower than observed values of 0.6 - 0.83. Better agreement was found for $BC/PM_{2.5}$ ratios with the modeled values of 0.05-0.33 as compared to the observation values of 0.05 – 0.28. AODEM (extended aerosol optical depth module) was used to calculate the total columnar aerosol optical depth (AOD) and BC AOD using the internal mixing assumption. The model AOD results were evaluated against the observed AOD by both AERONET and MODIS satellite in 10 countries in the domain. Our model results showed that the BC AOD contributed 7.5 – 12% of the total AOD, which was in the same ranges reported by other studies for places with intensive emissions. The Part 1 results (this study) is used in Part 2 (Permadi et al., 2017a) which calculates the regional aerosol direct radiative forcing under different emission reduction scenarios to explore potential co-benefits for air quality improvement, reduction in number of premature deaths and climate forcing mitigation in SEA in 2030.



## 1. Introduction

The Southeast Asia (SEA), with a large population and fast growing economy, is an important contributor to the emissions of air pollution and greenhouse gases in Asia (Streets et al., 2003; Zhang et al., 2009). The emissions of anthropogenic aerosol from Asia and specifically from SEA, are expected to rise in the near future due to the increase in the energy demand and rapid industrialization (Lawrence and Lelieveld, 2010; Ohara et al., 2007). High levels of fine particulate matter (PM with diameter less than 2.5 micron or $PM_{2.5}$), the most detrimental air pollutant from the health effect point (Janssen et al., 2011; WHO, 2012), are observed in many developing Asian cities with the annual average often exceeding the WHO guideline of 10 µg m$^{-3}$ by many times (Kim Oanh et al., 2006; Hopke et al., 2008). Being components of PM, e.g. $PM_{2.5}$ and $PM_{10}$ (size below 10 µm), black carbon (BC) and organic carbon (OC), have been monitored in some Asian cities and the results, although fragmented, showed considerably high levels (Kondo et al., 2009; Kim Oanh et al., 2006; Hopke et al., 2008). The fine particles and their precursors are also involved in the long distance transport hence causing the regional phenomenon such as Atmospheric Brown Clouds (ABC) (UNEP and C[4], 2002; Ramanathan et. al., 2001) and can affect the climate (UNEP-WMO, 2011). Globally, measures aim to reduce emissions of BC (and co-emitting pollutants) were shown to reduce the number of premature deaths and slow down the near future temperature increase and more benefits to be gained in Asia, where current emissions are high (UNEP-WMO, 2011; Shindell et al., 2012).

To comprehensively assess the co-benefits of emission reduction measures at regional scales, finer temporal and spatial resolutions of the modeling results are required. Several studies have been conducted for various Asian domains using the regional climate model (RCM) with chemistry (Nair et al., 2012) or chemical transport models (CTMs) with an additional aerosol optical module. Most of the Asian regional modeling studies mainly focused on the domains of East (Han et al., 2011; Park et al., 2011; Chen et al., 2013; Zhang et al., 2016)), South (Goto et al., 2011) and Continental East-Southeast Asia (Lin et al., 2014). These studies also highlighted several challenges for models to reproduce the ground-observed PM due to inaccurate emission inventory (EI), simulated meteorological fields and the extent of model representations (e.g., secondary organic aerosol formation, gas/particle partitioning, dry and wet deposition).

There is not yet detail modeling studies conducted for the SEA domain, especially the maritime SEA which includes Indonesia with its large biomass open burning (OB) emissions. For such a modeling effort, first reasonably accurate regional EI database should be prepared to generate input data. Several global and regional EI databases are available which also cover the SEA domain. These datasets have been developed using the activity data taken from several international data sources (Zhang et al., 2009; EC-JRC/PBL, 2010) or based on the results of large scale energy model (Streets et al., 2003). Efforts therefore should be put forward to update the SEA EI databases to generate the emission input data for the SEA regional modeling studies.

Our research used integrated EI and modeling tool to provide the spatial and seasonal distributions of aerosol species ($PM_{10}$, $PM_{2.5}$, and BC) in SEA for 2007 and the co-benefits (on air quality, health and climate forcing) of selected emission reduction measures for 2030. This paper (Part 1) presents the SEA emissions for the base year of 2007 and





the WRF-CHIMERE performance evaluation. CHIMERE (Menut et al., 2013 and references therein) was used to simulate three-dimensional (3D) aerosol concentrations in the domain using the meteorological fields generated by the Weather Research and Forecasting (WRF) model (Michalakes et al., 2004). The model results were evaluated using available ground-based measurements of $PM_{10}$, $PM_{2.5}$ and BC in several SEA cities. The extended aerosol optical depth (AOFD) module (AODEM), detailed in Landi and Curci (2011), was applied to calculate the total columnar AOD and BC AOD. The modeled total AOD was evaluated using the observed AOD from both ground based Aerosol Robotic Network (AERONET) and the Moderate Resolution Imaging Spectroradiometer (MODIS) satellite product. The results of Part 1 are used in the follow up study (Part 2) which investigated the potential co-benefits of various emission reduction measures implemented in Indonesia and Thailand on air quality improvement, number of premature death reduction and climate forcing mitigation in 2030 (Permadi et al., 2017a).

## 2. Methodology

### 2.1 Emission inventory and emission input data

The emissions from major anthropogenic sources (except for biomass open burning) in Indonesia, Thailand and Cambodia were developed using the EI framework given in the "Atmospheric Brown Cloud Emission Inventory Manual (ABC EIM)" (Shrestha et al., 2013) as summarized in Table 1. Detail EI methodology Indonesia was presented in Permadi et al. (2017b). For other countries in SEA, the emissions of $SO_2$, NOx, CO, VOC, $PM_{10}$, $PM_{2.5}$, BC and OC were taken from the available online gridded EI databases (grid size of 0.5° ~ 50 km) compiled by the Center for Global and Regional Environmental Research (CGRER) (Zhang et al., 2009). The gridded $CH_4$ and $NH_3$ emissions that were not included in CGRER, were taken from the global Emission Database for Global Atmospheric Research (EDGAR) (EC-JRC/PBL, 2010), with a grid resolution of 10 x 10 $km^2$.

The biomass open burning categories considered in this study included crop residue open burning (CROB) and forest fires (above-ground forest fires and peatland fires). The CROB emission (aerosol and trace gases) for Thailand for 2007 was taken from Kanabkaew and Kim Oanh (2011), and both CROB and above-ground forest fire emissions for Indonesia were from Permadi and Kim Oanh (2013) also for 2007. For other countries in the domain, the emissions from the above-ground forest fires were from Song et al. (2009) while those from CROB were from the Global Fire Emission Database v3 (GFED3) (Van der Werf et al., 2010). The emissions from peatland fires of all countries in the SEA domain were also taken from GFED3 (Van der Werf et al., 2010). The GFED3 database was developed using a combination of MODIS burned area and active fires which is believed to better detect the peatland fires than those used the MODIS burn scars area product of MCD45A1 for forest fires detection as quoted by Shi et al. (2014).

Biogenic emissions were calculated on-line in the CHIMERE model using the methodology described in Simpson et al. (1999) that considers seasons and vegetation cover types taken from the Global Land Cover Facility (GLCF) (http://glcf.umi aces.umd.edu) with a resolution of 1 x 1 $km^2$. CHIMERE incorporates the Model of Emissions of Gases and Aerosol from Nature (MEGAN) module (Guenther et al., 1995) for estimation of VOC and NOx emissions





from natural vegetation. Emissions from all considered sources were compiled for the base year of 2007 and were gridded to 0.25° x 0.25° (~ 30 x 30 km$^2$) for the modeling input.

The lumping of Non-Methane Volatile Organic Compounds (NMVOCs) emissions to the model species was done according to the MELCHIOR mechanism (Middleton et al., 1990). The aggregation produced the emissions of 33

species including trace gases and aerosol in unit of mole cm$^{-2}$ s$^{-1}$. Aerosol fluxes were also converted to the "molecule-like" units in the emission input data using a fictive molar mass equal to 100 g mole$^{-1}$ (Bessagnet et al., 2004).

### 2.2 Modeling domain

The choice of domain size and resolution affects the balance between the boundary and internal modeling forcing in the simulated concentrations (Seth and Giorgi, 1998). For this study, it is important that the defined domain allows

the transport of air pollutants by the monsoon circulation across SEA. Therefore, we set the domain to cover as much as possible the major upwind emission sources and to capture meteorological processes in the region of interest.

The SEA domain horizontally covered 9 (nine) countries of the Association of Southeast Asian Nations (ASEAN) and 3 provinces of southern China (Figure S1, supplementary information, SI). The WRF domain extended from middle part of Myanmar to the northern part of Australia covering 230 x 200 grids. The CHIMERE domain extended from

southern China (24ºN, 95ºE) to eastern parts of Indonesia (9ºS, 137ºE) consisting of 169 x 133 grids. The grid resolution of the WRF and CHIMERE was set to be the same, 0.25° x 0.25° (~30 x 30 km$^2$).

### 2.3 WRF and CHIMERE model configuration

WRF version 3.3 was used with lateral boundaries and initial meteorological conditions taken from the National Centers for Environmental Prediction (NCEP) final (FNL) global analyses that are available at 1º x 1º grid resolution

for every six hours (http://rda.ucar.edu/datasets/ds083.2/). WRF Pre-processing System (WPS) of geographical input data (i.e. land use, vegetation index, soil type, and albedo) was also obtained from the NCEP database. Totally, 28 vertical levels were simulated with the lowest level having the physical height of about 38 m. Analysis nudging was performed in the Planetary Boundary Layer (PBL) and other layers for wind components (u and v), temperature (T) and relative humidity (RH). Nudging coefficients were set for all parameters at 0.00005 s$^{-1}$. The time interval between

analyses was set at 360 minutes, which is equivalent to 6-hourly boundary input data used in our study. This analysis nudging was performed as it is suitable for a coarse resolution simulations (30 x 30km$^2$) to drive regional air quality models as it can improve the accuracy for the downscaled/nested fields (Dudhia, 2012; Bowden et al., 2012). Note that, due to the insufficiency of spatially distributed meteorological observations in the domain, the observation nudging was not performed.

In the WRF simulation, the following physics options were used: simple ice microphysics, Unified Noah land-surface model for land-surface scheme, Rapid Radiative Transfer Model (RRTM) and Dudhia schemes for long and short wave radiation, PBL parameterization scheme of Yonsei University (YU), and Kain-Fritsch (KF) scheme with deep



and shallow convection option for cumulus parameterization. These schemes were selected as they are suitable for mesoscale grid size and have been used in the previous studies in the world (Jankov et al., 2005; Osuri et al., 2012).

This study used CHIMERE version 2008c with the MELCHIOR 2 chemical mechanism that was adapted from the original European Monitoring and Evaluation Programme (EMEP) and consisted of around 120 reactions and 40

chemical species. The vertical profiles of updated reaction rates in MELCHIOR 2 have been developed using tabulated clear-sky photolysis rates taken from the Tropospheric Ultraviolet and Visible (TUV) model (EC4MACS, 2012). This version of CHIMERE has an aerosol module which consists of the total primary PM emission (BC, OC and other primary particles) and secondary inorganic PM species, such as nitrate, sulfate and ammonium, and secondary organic aerosol (SOA) (Bessagnet et al., 2004). CHIMERE applies the sectional approach to discretize particle size

distribution into a finite number of bins. The considered particles size range was from 40 nm to 10 μm that are distributed into 8 bins (0.039, 0.078, 0.156, 0.312, 0.625, 1.25, 2.5, 5, 10 μm) (Pere et al., 2011). Most of aerosol-related dynamic processes, such as condensation, coagulation, wet and dry deposition, adsorption and scavenging, are incorporated in the model (http://www.lmd.polytechnique.fr/chimere/). This version of CHIMERE only allows tropospheric simulations below 200 hPa (~12 km).

The vertical layers in this study were assigned to have 8 layers, from sigma level 0.999 (~20 m) to the sigma level ~0.5 (~5,500 m) or the 500 hPa pressure level. This upper limit was selected based on a suggestion that in the modeling of anthropogenic pollution, extending the vertical dimension beyond 500 hPa would not substantially change the modeled concentrations for the ground level (Menut et al., 2013). Monthly mean boundary conditions of gases and aerosol are the simulation results for a period of 1998-2002 by the Laboratoire de Météorologie Dynamique (LMDZ)

- Interaction avec la Chimie et les Aérosols (INCA) (Schulz et al. 2006) which are available at the CHIMERE website. Initial conditions of gases and aerosol concentrations in every grid were interpolated from the outputs of the global CTM of LMDZ-INCA simulation. One year simulation (January 1–December 31 2007) was performed by both WRF and CHIMERE with a spin up period of one week prior to the main simulation period.

**2.4 Aerosol Optical Depth Calculation**

A stand-alone post processing tool, namely AODEM, developed by Istituto di Scienze dell'Atmosfera e del Clima - Consiglio Nazionale delle Ricerche (ISAC-CNR) of Italy (Landi and Curci, 2011) was used to calculate optical parameters of AOD (extinction coefficients and single scattering albedo) using the 3D aerosol species mass concentration fields output of WRF/CHIMERE for different size bins. AODEM calculates 3D particle number concentrations from these mass concentrations and provides the extinction coefficients for each grid-cell assuming

the spherical shape of particles (http://people.isac.cnr.it/~landi/PhD.html). Three options of the aerosol mixing state were provided in AODEM: external, internal homogeneous and internal coated spheres. Aerosol optical properties are simulated by AODEM following the Mie theory (Bohren and Huffman, 1983) for the wavelength range from 340 nm to 1,640 nm. We selected the "aerosol internal mixing" option in the calculation because existing field measurements confirmed that aerosol is typically found in the internally mixed state (Lesins et al., 2002) largely due to coagulation

and growth of aerosol particles (Jacobson, 2000).





For calculation of optical aerosol properties, AODEM provides the particle number concentrations separately for five components: BC, OC, sea salt, dust, and secondary inorganics (nitrate, sulfate and ammonium). The AOD scattering was simulated using "brute force" by excluding BC in the simulation (Landi and Curci, 2011). BC AOD was calculated by substracting the AOD scattering from the total AOD.

**2.5 Model evaluation**

The evaluation of WRF outputs was done using observed data from eight (8) airport meteorological stations in 5 SEA countries that captured major sub-climate zones (upper, near-equator and lower latitude) in the domain. Hourly observations from all these airport stations in 2007 were obtained from http://weather.uwyo.edu/surface/meteorogram/. The statistical evaluation of WRF outputs was done using the criteria

provided by Emery et al. (2001) which includes the mean bias error (MBE), mean absolute gross error (MAGE) and root mean squared error (RMSE).

Only limited air pollution data was available in SEA for the model performance evaluation. This study collected the observed concentrations of aerosol (BC, OC, $PM_{2.5}$ and $PM_{10}$) and related gases from various sources. For examples, daily (24h) concentrations of $PM_{10}$, $PM_{2.5}$, BC, and OC in four SEA cities (i.e. Manila, Hanoi, Bandung, and Bangkok)

in 2007 were taken from the measurement data generated by the AIRPET project "Improving Air Quality in Asian Developing Countries" (Kim Oanh et al., 2006; Kim Oanh et al., 2014). Hourly BC and OC concentrations were taken from the measurement results of the Asian Pacific Network (APN) project at the AIT located in Pathumthani province of the Bangkok Metropolitan Region, Thailand (Kondo et al., 2009). Hourly $PM_{10}$ in Bangkok (Thailand), Kuala Lumpur (Malaysia) and Surabaya (Indonesia) in 2007 were collected from the respective national monitoring

networks. The statistical evaluation of simulated aerosol levels was done using mean fractional bias (MFB) and mean fractional error (MFE) (Boylan and Russel, 2006). Definitions of the statistical measures used in the model performance evaluation are given in Table S1, SI.

The monthly AERONET data for 2007 was downloaded from the National Aeronautics and Space Administration (NASA) website (http://aeronet.gsfc.nasa.gov/) for the evaluation of the modeled AOD The AERONET data was of

level 2 quality controlled, recorded at 10 AERONET stations (using Sun Photometer) listed in Table S2, SI. This provided AERONET dataset that has already been pre- and post-field calibrated with cloud screening and quality assurance. The selected 10 AERONET stations had more complete datasets in 2007 and they represent all sub-climate zones in the domain. Sun Photometer measures AERONET AOD at six different wavelengths (1020, 870, 675, 500, 440 and 380 nm). Therefore, to compare with the modeled AOD at 550 nm, the AERONET AOD at 500 nm were

converted to that 550 nm using a logarithmic interpolation (Chung et al., 2012).

For a qualitative evaluation of the spatial distributions of the modeled AOD, the satellite MODIS Terra and MODIS Aqua AOD were used. The consistency between the modeled AOD spatial distribution over the SEA domain for monthly MODIS AOD (level-3 data measured at 550 nm wavelength downloaded from





https://giovanni.sci.gsfc.nasa.gov/giovanni/#service=TmAvMp&starttime=&endtime=&variableFacets=dataFieldM
easurement%3AAerosol%20Optical%20Depth%3B).

## 3. Results and Discussion

### 3.1 Base year emissions

The obtained total national emission estimates of Indonesia, Thailand and Cambodia for 2007 were compared with
the existing regional EI databases of EDGAR for 2007 and CGRER for 2006. Table 2 shows a reasonable agreement
in the ranges of the estimates between the emission databases for three countries, Cambodia, Indonesia and Thailand.
Detail EI results for Indonesia was presented in Permadi et al. (2017b). There are certain discrepancies between the
databases that may be explained by several factors, including the inter-annual variations in forest fires, uncertainty in
activity data levels and EFs used, as well as the different coverage of the emission sources by the EI works.
Specifically, for the emission sources of $N_2O$, our EI for the three (3) countries did not cover the direct emissions from
cultivated soil (fertilized land) and the indirect $N_2O$ emissions from agriculture-related activities (microbial
nitrification and denitrification) hence resulting in lower $N_2O$ emission estimates. Similar reason may be used to
explain our lower estimate for $CH_4$ as compared to EDGAR for all 3 countries.

For Indonesia, our emission estimates were in between those of CGRER and EDGAR for a number of species. The
estimates for $PM_{10}$, $PM_{2.5}$ and BC actually agreed well between these databases while OC of CGRER appeared to be
higher. However, the $SO_2$ estimates differed a lot between the databases and our value was lower than others (mainly
for on-road transportation and industry) which may be attributed to a more bottom up approach used in our EI that
relied on actual S content used in the country and implementation of air pollution control devices (Permadi et al.,
2017b). For example, our $SO_2$ estimate for the power plants of Indonesia was 300 Gg $y^{-1}$ which was more comparable
with the CGRER estimate (409 Gg $y^{-1}$) but much lower than the EDGAR estimate (1,000 Gg $y^{-1}$). The most striking
difference was for the $CO_2$ emission that showed a much higher value by EDGAR. However, this could be clearly
explained by the inclusion of two major sources in the EDGAR dataset: i) the forest fire post burn decay (698,000 Gg
$y^{-1}$) and ii) decay of drained peatland (504,000 Gg $y^{-1}$). If these two sources are excluded from the EDGAR results,
the $CO_2$ estimates of all 3 databases are similar for Indonesia. The emission estimates for Thailand and Cambodia also
showed reasonable agreements between the available datasets. However, EDGAR appeared to provide higher BC
emissions for these two countries while CGRER provided higher OC emission for Thailand, i.e. similar to Indonesia
case above.

The emission shares by source category for the three countries are presented in Figure S2, SI. The emissions of aerosol
species ($PM_{10}$, $PM_{2.5}$, BC and OC) were mainly from the residential and commercial combustion in Indonesia (43-
80%) and Cambodia (55-78%) while for Thailand the biomass OB (forest fire and crop residue) emissions were
dominant, i.e. 31-74%. For $SO_2$, the emission in Indonesia was mainly contributed by the transport sector (36%) and
thermal power plants (33%) but the industry was the main contributor in both Thailand (66%) and Cambodia (33%).





For NOx, the total emission in Indonesia was dominated by the fugitive emissions from oil and gas operation (44%), in Thailand by power plant (34%) and in Cambodia by forest fires (60%). The total emission of $NH_3$, an important precursor for $PM_{2.5}$, in all three countries was mainly from the manure management and fertilizer application (others), i.e. 63% for Indonesia, 75% for Thailand and 78% for Cambodia.

The emissions from other SEA countries and from the non-SEA part (southern part of China) of the domain used in our modeling study are also included in Table 3. The emissions from the southern China had high shares in the total emissions from the modeling domain. It is seen that Indonesia and Thailand were collectively the largest emitters of all pollutants, sharing of 25-66% of 2007 SEA emissions and 17-44% of the modeling domain emissions. Thus, emission reduction measures implemented for these 2 countries are expected to contribute remarkably to the air quality

improvement in the region which will be analyzed in the companying paper (Part 2). The spatial distributions of the annual average emissions of BC and CO at 0.25° x 0.25° (~30 x 30 km²) resolution over are presented in Figure 1 that showed higher emission intensity over large urban areas in the domain.

### 3.2 WRF model results and evaluation

The WRF hourly outputs, including surface temperature (T), relative humidity (RH) and wind speed (WS) for 2007

were compared with the observed data at 8 international meteorological stations in 5 SEA countries (Table 3). The comparison was done for two seasons, 3 months, 1 January – 31 March, to represent the dry season in the continental SEA (but the wet season in Indonesia) and 3 months, 1 August – 31 October, to represent the wet season in the continental SEA (but the dry season in Indonesia). The time series of daily average modeled vs. observed meteorological parameters, as shown in Figure S3a-S3b, SI, showed that the model appeared to reasonably reproduce

all parameters for the considered stations. In general, the model performance for temperature and WS simulations at all the stations was better than for RH during both periods.

The statistical performance evaluation, based on hourly simulated, against the MBE, MAGE and RMSE criteria is given in Table 3. MBE for the January-March period range was -0.4 – 10.2 °C for T, -0.3 – 2.7 m s⁻¹ for WS, and -13 – 29.6% for RH. The corresponding range obtained for the August-October period was -0.6 – 2.1 °C, -0.6 - 2.1 m s⁻¹

and -6.7 – 5.6%. Other statistical measures of MAGE and RMSE varied between the stations and the deviations from the suggested criteria were generally small. This suggested a relatively good model performance of WRF for both dry and rainy seasons. Overall, for the stations located in the north latitudes (above the equator line), the model performed better in the wet season (August – October) while for those located near and lower than the equator line the model performance was equally good for both dry and wet seasons. The discrepancy between model results and observations

was perhaps partly due to the fact that the domain cover some regions, such as the Indonesian maritime – continent, that are principally characterized by active convection with a frequent presence of deep convection. These local processes, e.g. deep convection, are difficult to simulate using the mesoscale meteorological model of WRF with a rather coarse resolution (0.25° ~ 30 x 30 km²) used in this SEA modeling study. Therefore, finer resolutions are required to capture the dynamical processes undergoing at smaller scales. Different physics options may be required

for sub-region domains to capture the processes which should be done in future studies. In addition, a certain





discrepancy is always expected because the model provided a grid average value, i.e. one for a 30 x 30 km² cell, while the observation is point based at individual stations.

### 3.3 CHIMERE model results and evaluation

Aerosol simulation always presents a big challenge due to the complex multiphase chemistry and transport processes. Lack of ground monitoring data of aerosol in the SEA region is an obstacle to a comprehensive model performance evaluation. For model performance evaluation, the CHIMERE results of $PM_{10}$, $PM_{2.5}$, BC and the ratios of $PM_{2.5}/PM_{10}$ and BC/PM are discussed when comparing with available observed data in the domain in 2007.

### 3.3.1 $PM_{10}$

The daily (24h) modeled $PM_{10}$ concentrations were estimated using the hourly data and the results were compared with the data gathered from the governmental monitoring networks that are available in three big cities of SEA, (i.e. 1 station in Kuala Lumpur, 2 stations in Bangkok, and 1 station in Surabaya). Note that the same two periods, as for WRF evaluation above, were used to represent dry and rainy season for both northern and southern parts of the equator. Overall, model results ranged from near zero to 85 µg m⁻³ while the observations ranged from 5 to 90 µg m⁻³ at the three cities. The period average of modeled $PM_{10}$ in the three cities ranged from 21.7 – 29.2 µg m⁻³ while the corresponding observations ranged from 25.9 to 45.2 µg m⁻³ (Figure 2).

Scatter plots of daily average observed and modeled values are presented in Figure 2 showed that the model appeared to reasonably capture the range of 24h $PM_{10}$ in the cities but it showed non-linearity correlation. The model underestimated the low observed values at the Kuala Lumpur station (one station), i.e. the observed levels were 30-60 µg m⁻³ while the modeled were fluctuating from near zero to about 60 µg m⁻³. A better agreement in the range of 24h $PM_{10}$ was shown for Surabaya, i.e. both were from 5 µg m⁻³ to 85 µg m⁻³, but the linear correlation was still quite low. For Bangkok, the modeled 24h $PM_{10}$ ranged from 10 - 60 µg m⁻³ while the upper limit of the observed values was 90 µg m⁻³. It is noted that although the ranges of the modeled 24h $PM_{10}$ were comparable with the observed ranges but the correlations were not clear for all three cities.

The reason for the discrepancy in the day to day variations between the modeled and observed 24h $PM_{10}$ values could be due attributed to the less accuracy of the temporal variations of the emission input data, as well as due to the coarse resolution of the model which may not be able to represent the weather variables in a convection-dominated climate. It is always challenging to compare the regional scale modeling results obtained for a coarse resolution (i.e. 30 x 30 km²) with the point-based observations especially in complex mixed urban areas. Lack of systematic monitoring data for $PM_{10}$ in rural sites of the domain during the modeling periods prevented from making a more comprehensive model performance evaluation. The statistical evaluation showed that in all 3 cities, the MFB and MFE values for 24h $PM_{10}$ (totally 179 data points for each city) were within the suggested criteria (Table 4). The MFB values in Bangkok, Kuala Lumpur and Surabaya were -53%, -56% and -9%, respectively, i.e. lower than the criteria of ≤±60%. The MFE values in Bangkok, Kuala Lumpur and Surabaya were 55%, 56% and 18%, respectively, which were also well within the criteria of ≤+75%.





The simulated monthly average of $PM_{10}$ in Kuala Lumpur and Bangkok, were consistently lower than the observed values in all months (Figure 2) which should be expected in principle due to the grid averaging of the model results. For Surabaya, however, the model simulated monthly $PM_{10}$ values were higher than the observed during the period of January – March 2007 but lower than the observed for the period of August – October. The discrepancy between the modeled and observed $PM_{10}$ for this city may be caused by the limited monitoring data availability along with other uncertainties associated with the modeling results which should be addressed in future studies at the urban scale.

### 3.3.2 $PM_{2.5}$

Only some fragmented measurement data of $PM_{2.5}$ was available in the domain in 2007 for the model evaluation. This study used the 24h $PM_{2.5}$ data monitored in the SEA cities of Bandung, Bangkok, Hanoi and Manila, under the AIRPET project (Kim Oanh et al., 2006; Kim Oanh et al., 2014). The observation data were only available for some specific periods in 2007 at different sites hence the modeled results were extracted for the corresponding periods for comparison. The observed sites were the mixed sites which were influenced by typical emission sources in the respective cities. The AIT site, located of about 650 m away from a heavily travelled road, represented a sub-urban site with the influences of emissions from traffic and open burning of rice straw (Kim Oanh et al. 2009). Thuong Dinh (TD) of Hanoi was a mixed urban site influenced by traffic and residential combustion among other sources (Hai and Kim Oanh, 2013). Both Tegalega (TG) located in Bandung, Indonesia and Manila observatory (MO) in Manila, Philippines were urban mixed sites with strong influence of traffic and other urban typical sources. The data therefore represents different periods of the year and different urban characteristic sites hence was not meant to compare the levels between the cities but only for model performance evaluation.

Overall, the available observed 24 $PM_{2.5}$ data in four AIRPET cities ranged from 4 to 120 $\mu g\ m^{-3}$ while the modeled values for the same data periods ranged from 5 to 64 $\mu g\ m^{-3}$. The average levels of the observed $PM_{2.5}$ over all the data periods ranged from 35 to 43 $\mu g\ m^{-3}$ as compared to the modeled, i.e. from 9.7 to 21 $\mu g m^{-3}$. Scatter plots of observed and modeled 24h $PM_{2.5}$ at four AIRPET stations (Figure 3) clearly showed that the model underestimated 24h $PM_{2.5}$ in all stations. In the mixed polluted urban site in Bandung (TG), modeled 24h $PM_{2.5}$ was within a range of 11-33 $\mu g\ m^{-3}$ while the observed were 27-69 $\mu g\ m^{-3}$. In the TD urban site in Hanoi (close to a busy road), the simulated 24h $PM_{2.5}$ were 5-64 $\mu g\ m^{-3}$ as compared to the observed of 20-120 $\mu g\ m^{-3}$. In the mixed urban site of MO in Manila the simulated 24h $PM_{2.5}$ were 6-37 $\mu g\ m^{-3}$ as compared to the observed range of 4-55 $\mu g\ m^{-3}$. As discussed above, the four selected AIRPET sites were located quite close to heavily travelled roads (although were not directly at roadside) hence the local traffic emissions could directly affect the monitored pollution levels. This may be an important reason for the discrepancy between the monitored levels and the simulated grid average values. In addition, the observed data points were quite limited for 2007 (≤30 at each site) hence were not sufficient for the statistical model performance evaluation. The $PM_{2.5}$ monitoring efforts should be enhanced to characterize the pollution in SEA and also provide sufficient data points for the model evaluation.



### 3.3.3 Black carbon

For the model evaluation purpose we used available measurements in the previous projects for SEA. The 24h BC measured by the optical method available at several SEA sites under the AIRPET project (Kim Oanh et al., 2014). The hourly-based EC (elemental carbon by a Sunset analyzer) measurements, available from the APN project (Kondo et al., 2009) for the AIT site (sub-urban) which were used to calculate 24h BC levels. The model performance evaluation was done using 24h BC data of both APN and AIRPET projects.

The APN hourly EC dataset for the AIT site was available for both dry and wet seasons, from March to December 2007. The hourly EC (Sunset) and hourly BC (optical method, using the continuous soot monitoring system or COSMOS) measured simultaneously by the APN project at AIT were found to have a strong linear correlation (Kim Oanh et al., 2009). Therefore, we used the observed Sunset EC to compare with the modeled output of BC. Figure 4 presents the time series of the modeled and observed 24h BC for the AIT site. The modeled 24h BC was from 1.0 to 10 $\mu g\ m^{-3}$ that is comparable with the observed range from 0.8 – 10 $\mu g\ m^{-3}$. However, correlation between the modeled and observed BC shown in the scatter plot was not clear. The discrepancy between the modeled and observed BC seen in the time series may principally be due to the gridded average of the model output as compared to the point-based measurement. Higher BC levels measured at the AIT site were contributed by multiple local sources, such as nearby highway traffic activity and biomass open burning (of rice straw) that occurred more intensively during the dry season (December). However, these sources especially small scale rice straw field burning activity may not be well represented spatially by the EI input data made for a large resolution (30 x 30 $km^2$). Three (3) statistical measures of MBE, MFB and MFE were considered for the model performance evaluation in the BC simulation at the AIT site (Table 4). The MFB and MFE values were -24% and 49%, respectively, which all meet the suggested criteria (for PM). The MBE value was -0.12 $\mu g\ m^{-3}$ for AIT site which showed that the model somewhat underestimated the observed BC values but there is no criteria MBE available for PM for comparison.

The 24h BC (optically) measured on the 24h $PM_{2.5}$ sampled filters collected in the same locations of $PM_{2.5}$ measurements in SEA under the AIRPET project (Kim Oanh et al., 2006; Kim Oanh et al., 2014) were compared with the 24h modeled BC extracted for the sites and dates of 2007. Figure 5 shows that the modeled 24h BC were lower than the observed at all the sites. The ranges of observed values and the modeled values were in somewhat better agreement for the AIT site and MO Manila site than the other 2 sites. At AIT, the observed BC values was 1.3 – 3.4 $\mu g\ m^{-3}$ (January, February and May) were higher but quite comparable to the modeled range of 0.5 – 1.8 $\mu g\ m^{-3}$. At MO, the observed 24h BC was 7 – 13 $\mu g\ m^{-3}$ (January and February) was quite close to the modeled 24h BC of 4.2 - 13 $\mu g\ m^{-3}$. More discrepancies were found for the Bandung site with the observed 24h BC values ranged from 4.2 to 9.8 $\mu g\ m^{-3}$ (July 2007) as compared to the modeled values of 1.3 – 3.2 $\mu g\ m^{-3}$. Similarly, the observed BC values at the mixed site of TD, Hanoi ranged from 12 to 23 $\mu g\ m^{-3}$ (January 2007), much higher than the modeled values of 1-7 $\mu g\ m^{-3}$. The effects of local sources, especially traffic emissions, at the quoted sites should be a main cause of the discrepancies when compared to the grid average modeled BC with the observed values. The limited measurement




data available prevented from a more comprehensive model performance evaluation. Note that due to the limited measurement data points, a statistical performance evaluation was not conducted for the BC simulation.

### 3.3.4 Ratios between fine and coarse PM, and between BC and PM

In fact, $PM_{2.5}$ mass is principally contributed by both local combustion sources and secondary particles formation by chemical reactions in the atmosphere. The gaseous precursors of NOx, SOx and VOCs for the $PM_{2.5}$ formation may be of both local and long range transported origins. The coarse fraction ($PM_{10-2.5}$) would mainly consist of primary particles of the geological origin (Chow et al., 1998), and these are mainly contributed by local sources of soil, road dust and also construction activities (Hai and Kim Oanh, 2013). Due to its formation process as well as the ability to participate in the regional transportation, the fine particles ($PM_{2.5}$) are more uniformly distributed in an urban area than the coarser particles. The $PM_{2.5}/PM_{10}$ ratios could provide some information of the dominance of local sources of $PM_{2.5}$. We compare the $PM_{2.5}/PM_{10}$ ratios based on the modeled 24h $PM_{2.5}$ and 24h $PM_{10}$ ($PM_{10} = PM_{2.5} + PM_{10-2.5}$) and those computed the observed PM data available at the four (4) AIRPET monitoring sites discussed above. Overall, the modeled $PM_{2.5}/PM_{10}$ ratios ranged from 0.47 to 0.59 while the observed values were higher, 0.6 - 0.83. More pronounced difference was for TD of Hanoi, i.e. 0.74 observed vs. 0.47 modeled, and for TG of Bandung, 0.83 observed vs. 0.55 modeled. Better agreements were obtained for MO of Manila, 0.61 observed vs. 0.47 modeled, and the AIT site, 0.6 observed vs. 0.59 modeled. The urban mixed sites of TD in Hanoi and TG in Bandung located in the traffic areas hence higher contributions of the primary $PM_{2.5}$ emitted from traffic may be seen better for TD and TG sites to the total measured $PM_{10}$, as compared of MO in Manila and AIT sites. However, to evaluate the variations in the $PM_{2.5}/PM_{10}$ ratios, contributions of various sources of the coarse particles, such as road dust and construction dust, should be further analyzed.

BC is emitted directly from the combustion sources with higher fractions in PM emitted from the diesel exhaust (Kim Oanh et al., 2010) and lower fractions from biomass open burning (Kim Oanh et al., 2011). Hence the ratio of $BC/PM_{2.5}$, for example, can infer the contribution of the primary particles from these combustion activities. $BC/PM_{2.5}$ and $BC/PM_{10}$ ratios were calculated using the observed 24h data at four AIRPET sites. Modeled $BC/PM_{2.5}$ ratios ranged from 0.05 to 0.33 as compared to the observed ratios of 0.05 – 0.28. For $BC/PM_{10}$, the modeled values ranged from 0.03 to 0.16 while the observed values ranged from 0.034 to 0.17. Observed $BC/PM_{2.5}$ ratios were higher than the modeled values at TG of Bandung (0.16 vs. 0.1) and AIT (0.055 vs. 0.05) sites. In TD of Hanoi and MO of Manila, the observed ratios (0.22 and 0.23) were lower than the modeled (0.28 and 0.33). As for $BC/PM_{10}$, the observed ratios at three (3) AIRPET sites of TG, TD and AIT (0.13, 0.17, and 0.034) were higher than the modeled values (0.06, 0.13 and 0.03) while for MO of Manila the opposite was shown with a lower observed (0.14) as compared to the modeled (0.16) value. The simulated BC/PM ratio was the highest in TD of Hanoi, 22% of $PM_{2.5}$ and 17% of $PM_{10}$, during the dry period of January – February 2007 which confirmed the strong influence of traffic emission at this site.

As seen in the statistical model evaluation, a negative MB was obtained for $PM_{10}$, -3 to -17, and BC, -0.12 (BC) at all sites (not enough data for statistical evaluation of $PM_{2.5}$) which showed an underestimation of $PM_{10}$ and BC


concentrations by the model at all sites. This may be explained by the coarse resolution (30 x 30 km$^2$) of emission input data which could adequately represent the spatial distributions of local sources of a smaller scale such as road traffic. These local sources, for example road traffic and residential cooking, affect PM measured at all sites hence affecting the $PM_{2.5}/PM_{10}$ and BC/PM ratios. The road/soil dust emission contribute more to $PM_{10-2.5}$ hence lowering

$PM_{2.5}/PM_{10}$ ratios in urban areas but this coarse fraction of PM emission was not included in our emission input file. Thus, in future studies these sources should be included to improve the $PM_{10}$ outputs from the models.

### 3.4 Spatial distribution of modeled monthly $PM_{10}$, $PM_{2.5}$ and BC

Spatial distributions of the modeled monthly average $PM_{10}$, $PM_{2.5}$ and BC are presented in Figure 6 for January,

August and November while those of the respective annual averages are presented in Figure S4, SI. The highest monthly average concentrations of $PM_{10}$ in January, August and November 2007 simulated in the domain (one value for the whole domain) were 69, 58, and 44 µg m$^{-3}$ while corresponding values of $PM_{2.5}$ were 40, 37 and 27 µg m$^{-3}$, respectively. The simulated maximum monthly average BC concentration in the domain was higher in January (8.2 µg m$^{-3}$) as compared to August (6.8 µg m$^{-3}$) and November (6.2 µg m$^{-3}$).

The simulated highest hourly $PM_{10}$ in the considered months of January, August and November 2007 were 325, 245 and 164 µg m$^{-3}$, respectively, while the $PM_{2.5}$ corresponding values were and 188, 150 and 99 µg m$^{-3}$. The highest values of simulated annual average in the domain for $PM_{10}$ and $PM_{2.5}$ were 51 and 32 µg m$^{-3}$, respectively. The maximum simulated annual average in the domain for BC was 6 µg m$^{-3}$. A summary of the simulated pollutant levels in the domain is presented in Table S3, SI.

For all considered pollutants over the domain, higher concentrations were observed over East Java, Indonesia, particularly over Surabaya city, which show the effects of emission from residential and traffic in the city and surrounding satellite cities as well as the crop residue OB (Permadi and Kim Oanh, 2013; Permadi et al., 2017b). High concentrations were consistently observed in several places in Indonesia including Java Island, West Sumatera (Padang), and West Kalimantan (Pontianak), and over Bangkok, Thailand. Large hotspots but with lower

concentrations were also observed over the Southern China, over Hanoi and Ho Chi Minh of Vietnam which can be largely explained by the influence of the local sources (Figure 6).

On the regional scale, the monsoon circulation plays an important role in transporting PM from the emission source regions to other parts of the domain. In the dry months, higher emissions of biomass OB are expected hence higher

concentrations of PM should be seen in the region near and downwind sources. Accordingly, at the northern part of the domain, higher PM levels were found in January-April while at the southern part of the domain higher concentrations were found during the period of April-August. In January at the upper latitude, northeast monsoon transports pollutants from the source regions to the southwest direction while at the lower latitude (Indonesia) the plume moved to the northeast/east direction. In August and November at the lower latitudes the plume of PM moved





northwesterly/westerly direction while in the upper latitude southwest monsoon brought the pollutants to the northeast direction (Figure 6).

In August and November, the dry months in the Southern domain, the $PM_{10}$ and $PM_{2.5}$ plumes showing the effects of biomass OB (crop residue and forest fire) emissions in Indonesia originated in Riau province (Sumatera Island) and western and southern parts of Borneo Island, were seen clearly moving northeast-ward. In the dry season month in northern domain of January, the plumes of $PM_{10}$ and $PM_{2.5}$ intensified by biomass OB in the central and northern parts of Thailand were shown moving southwest-ward. BC plumes were generally seen originated from big cities in the domain showing a significant influence of the fossil fuel combustion emission, specifically traffic, and other urban

activities for all months of the year. During the dry period, BC plumes from the areas that have intensive biomass OB emissions were not as clearly seen as the PM plumes and this maybe because biomass OB contributed more to OC rather than BC emissions.

### 3.5    Aerosol optical depth

Both total AOD and BC AOD were considered for the model evaluation. The monthly average of the total columnar

AOD (scattering and absorbing), at the wave length of 550 nm, was produced from the AODEM simulation for 2007. The simulated monthly AOD data was compared with the monthly Terra-MODIS AOD, also at 550 nm, retrieved from the NASA website. Figure 7 showed that the modeled AOD was lower than the MODIS observed for example in January, the maximum AOD simulated for the Southern China part of the domain was about 0.36 as compared to the MODIS AOD of 0.42-0.58. In the same month, the modeled AOD values over the Java Island of Indonesia were

0.072-0.28 while the MODIS AOD was 0.26-0.42. In April, the model results over the Southern China was 0.25-0.75 while the observed MODIS AOD was 0.42-0.90. Near the border between Myanmar and Bangladesh (northwest corner of the domain), the modeled AOD and the observed MODIS AOD was similar 0.74-0.75. However, the modeled AOD values over the Java Island in April were higher, i.e. 0.02-1.0, than the observed MODIS AOD of 0.26-0.42. The simulated hourly maximum and monthly average $PM_{10}$ and $PM_{2.5}$ concentrations over Java Island were the

highest throughout the year in April which may be due to higher emissions from residential combustion and traffic during the big Moslem Festival in the country (Permadi et al., 2017b).

In December, a hotspot with the maximum AOD of 0.8 was observed by MODIS in the Eastern part of Java Island which is well above the model result for the grid of 0.37. The model was not able to capture hotspots over the Southern China mainland in December but produced high AOD values over the ocean part. The results for August and

November both showed some significant underestimation of AOD as compared to the MODIS observed values. There are several reasons for these discrepancies, including the temporal and spatial inconsistency in the observed and modeled values used for comparison. For example, the Terra MODIS satellite daily passed a region for a particular time (i.e. 13:30) hence giving only a snapshot of the value while the model provided the hourly average for 13:00 – 14:00 hence there is certain inconsistency in the monthly averages derived from these two datasets. Different spatial

resolutions of modeled AOD (30 x 30 $km^2$) and MODIS AOD (10 x 10 $km^2$) can be another reason. In addition, the



natural sources of aerosol, such as wind-blown dust, were not included in our emission input data hence the model would produce lower AOD (as well as PM$_{10}$) values. Overall, this qualitative analysis of the modeled vs. MODIS AOD provided some insight into the regional distributions although more efforts are still required for the model evaluation.

Simulated monthly AOD values were also compared with the observed data retrieved from 10 AERONET stations located in the domain, i.e. in Vietnam, Singapore, Hong Kong, Taipei, Thailand and Indonesia, which also showed lower simulated AOD values than the AERONET observed (Figure 8). The model appeared to capture better seasonal variability in most of the stations in Thailand and Vietnam than the stations in other countries. The strong seasonal variation of aerosol in SEA, largely caused by the biomass open burning, creates a huge challenge for models to

reproduce. At Puspitek Serpong (Indonesia) where emissions of urban activities from the capital city of Jakarta would dominate, the high AOD in October was reasonably captured by the model. The seasonal variation in the emission input file would need to be further refined to improve the situation.

The BC AOD (absorbing) was calculated as the difference between the total AOD (scattering + absorbing) and the scattering AOD following the same method presented in Landi and Curci (2011). The spatial distribution of monthly

average BC AOD is presented in Figure S5, SI. In January, the dispersion plumes of high BC AOD spreading over in the Southern China (maximum AOD of 0.027) and eastern part of Indonesia (maximum 0.018) which shared 7.5-10% of the total AOD of 0.36 and 0.18, respectively, in these areas. In April, the highest value of the modeled BC AOD was seen over Surabaya (East Java province, Indonesia) with the range of 0.051-0.078, followed by relatively high values over Hong Kong and Shenzhen of 0.06-0.069. The contributions of the BC AOD to the total AOD in Surabaya,

Hong Kong and Bangladesh were 9% (of 0.89), 11% (of 0.6) and 12% (of 0.54), respectively.

In other months, the highest monthly average BC AOD was shown in different parts of the domain ranging between 0.015-0.027 while the total AOD were 0.18-0.36, hence the shares of BC AOD in the total AOD was 7.5-8.6%. Our BC AOD contributions to the total AOD were higher than the reported global average value of 3% (Reddy et al., 2005), but in the same range of those reported for different regions with intensive emission sources. The relative

contribution of BC to total AOD has been reported to depend on wavelength, i.e. increasing with decreasing wavelength, and on the dominant emission sources. For example, measurements showed typical contributions of around 12% under the influence of natural dust (Chiapello et al., 1999) and around 5-12% if biomass OB is dominant (Eck et al., 1999; Dubovik et al., 2002). The modeled BC AOD serves as input to estimate BC direct radiative forcing of anthropogenic emissions for the SEA domain which will be analyzed in our companying paper (Permadi et al.,

2017a).





## 4 Summary and conclusions

This study developed and evaluated the EI databases for Indonesia, Thailand and Cambodia for 2007. The results were compiled with the existing CGRER and EDGAR emission datasets to generate the emission input data of the entire SEA domain for regional WRF/CHIMERE modeling. Our EI results for the three countries were comparable to other existing databases and the differences are explained mainly by the differences in the sources covered by different EI works. The BC emissions were mainly from residential and commercial combustion in Indonesia (71%) and Cambodia (70%) but was dominated by biomass OB emissions in Thailand (31%).

The model performance for 2007 was evaluated using the hourly and daily observed data in the SEA domain. The WRF model outputs were in good agreement with the observed data at 8 international airport stations in Indonesia, Thailand, Vietnam, Cambodia and Philippines. The WRF/CHIMERE satisfactorily reproduced the aerosol species of $PM_{10}$, $PM_{2.5}$ and BC in terms of the spatial distributions and seasonal variations. The statistical evaluation was conducted for 24h $PM_{10}$ and 24h BC which had sufficient observed data points for the analyses. The modeled 24h $PM_{10}$ in three cities (Thailand, Malaysia, and Indonesia) had the MFB and MFE values met the suggested criteria. Similarly, the modeled 24h BC values met the MFB and MFE criteria for PM when compared the observed data at a sub-urban site in Thailand (AIT).

The $PM_{2.5}/PM_{10}$ ratios calculated from the modeled outputs were lower than those estimated from the observed data at four AIRPET sites and this would imply a necessity of further improvement of the PM speciation of the emission input data. The modeled $BC/PM_{2.5}$ ratios were in compatible range (0.05 - 0.33) with the observed values (0.05 – 0.28) and were lower in two sites (AIT and Bandung) but higher in the others (Hanoi and Manila). The modeled $BC/PM_{10}$ ratios ranged between 0.03 - 0.16 which were comparable to while the observed values range (0.034 - 0.17). Lack of systematic observed BC data prevented from a more comprehensive model performance evaluation. Nevertheless, further improvement of the EI for primary aerosol, especially the PM speciation of major sources, as well as inclusion of un-paved road and wind-blown dust emissions are highly required.

The spatial distributions of the columnar total AOD estimated for the WRF/CHIMERE output PM concentrations using AODEM were comparable with the observed (MODIS and AERONET) in 2007. In particular, exclusion of the un-paved road and wind-blown dust emissions (coarse particles) from the emission input in this study was a reason for the discrepancy in the modeled and observed total AOD which may underestimate the coarse PM concentrations. The lower values of aerosol species simulated by the model were explained by the grid averaging effects: WRF/CHIMERE had a larger grid of 30 km, as compared to MODIS AOD of 10 km, while AERONET is actually point based. Therefore, the spatial distribution of local sources of a smaller size can not be captured well by WRF/CHIMERE.

The spatial distribution patterns of the modeled aerosol species in the domain may be explained by the intensive biomass OB emissions. The plumes of $PM_{10}$ and $PM_{2.5}$ originated from Sumatera and Borneo Island of Indonesia in August-November, and from Central and Northern Thailand during January-April, which coincided with the dry



months in the respective areas with subsequent more biomass OB. Spatial distributions of BC showed the influence of the traffic emission and residential combustion in big SEA cities. Based on the model results, the contribution of BC AOD to total AOD in the domain was around 7.5 – 12% which is consistent with the literature reported values for intensive emission areas.

The EI data and WRF/CHIMERE performance for 2007 were satisfactory in terms of reproduction of the key aerosol species in the domain. In the accompanying paper (Permadi et al., 2017a) we present the WRF/CHIMERE simulation results for PM and BC for the SEA domain in the business as usual emission scenario (BAU2030) and in the emission reduction scenario (RED2030) to quantify potential co-benefits on the air quality improvement, reducing number of premature deaths, and radiative forcing mitigation in Southeast Asia.

**Acknowledgements**

This research was financially supported by the French government under the Asian Institute of Technology (AIT)/Sustainable Development in the Context of Climate Change (SDCC) – France Network cooperation and by the United States Agency for International Development (USAID) under the PEER-SEA (Partnerships for Enhanced Engagement in Research for Southeast Asia) Research project. We thank APN and AIRPET projects for making BC and PM data available at four sites for the model evaluation. AERONET principal investigators are highly acknowledged for making data available on-line. Prof. Talib Latif from University Kebangsaan Malaysia (UKM) is highly acknowledged for sharing the information on $PM_{10}$ monitoring data in Kuala Lumpur, Malaysia; Dr Tony C Landi (Instituto di Scienze dell'Atmosfera e del Clima (ISAC), Italy) and Prof Gabrielle Curci (University of L'Aquila, Italy) are thanked for making the source code and users guide of AODEM available in the website.

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





**Table caption**

5    **Table 1: Summary of activity data level from different emission sources in 3 countries**
**Table2: EI results for base year in comparison with the existing regional EI datasets (Gg y$^{-1}$)**
**Table 3: Statistical parameters for WRF Model Performance evaluation for two periods**
**Table 4: Statistical parameters for CHIMERE model performance (PM$_{10}$ and BC) evaluation**





**Table 1: Summary of activity data level from different emission sources in 3 countries.**

| Sectors | Types of activity data | Activity data | | |
|---|---|---|---|---|
| | | Indonesia | Thailand | Cambodia |
| Power generation | Fuel consumption (Mt y$^{-1}$): | | | |
| | - Coal | 23.4 | 20.5 | - |
| | - Natural gas | 3.2 | 29.8 | - |
| | - Fuel oil | 9.4 | 0.75 | 0.62 |
| | - Biomass | 6.3 | - | - |
| Manufacturing industry | Fuel consumption ( Mt y$^{-1}$) | | | |
| | - Coal | 5.4 | 12.3 | - |
| | - Gasoline | 0.34 | 0.013 | - |
| | - Fuel oil | 1.8 | 2.4 | 0.52 |
| | - Biomass | - | 20.7 | - |
| On road transport | Number of registered vehicle (Million y$^{-1}$) | 48 | 26 | 1.9 |
| Air traffic | LTO (*1000/year) | 344 | 555 | 39 |
| Residential & commercial | Fuel consumption (Mt y$^{-1}$) | | | |
| | - Coal | 0.028 | - | - |
| | - Wood | 100.5 | 7.6 | 0.4 |
| | - Kerosene | 7.3 | 0.13 | 0.003 |
| | - LPG | 1 | 1.15 | 0.005 |
| | - Charcoal | 20.4 | 3.9 | 0.042 |
| | - Other biomass | - | 0.14 | - |
| Fugitive emission from fuel | - Coke production (Kt y$^{-1}$) | 182 | - | - |
| | - Gas production (Tg y$^{-1}$) | 8,654 | 31.2 | - |
| | - Oil production (Tg y$^{-1}$) | 29 | 6.2 | - |
| | - Gasoline distributed (Mt y$^{-1}$) | 13.7 | 5.4 | - |
| Agro residue open burning | Total dry crop residue openly burned (Mt y$^{-1}$) | 43.5 | 18.2 | 4.3 |
| Forest fire | Total forest area burned, including peatland fire (ha y$^{-1}$) | 545,881 | 1,851,850 | 98,761 |
| Solid waste open burning | Total dry solid waste burned (Mt y$^{-1}$) | 1.26 | 0.28 | 0.175 |
| Agriculture related activities | - Total number of livestock population (head, *10$^6$) | 1,359 | 328 | 22.3 |
| | - Fertilizer consumption (Mt y$^{-1}$) | 6.8 | 3.6 | - |
| Solvent and product use | - Paint (Kt y$^{-1}$ of paint) | 606 | ne | ne |
| | - Degrease (t y$^{-1}$ of solvent consumed) | 103 | | |
| | - Chemicals (Kt y$^{-1}$ of products) | 1,269 | | |
| | - Other products use (i.e. ink, domestic solvent, glue and adhesives) (Kt of products) | 161 | | |



**Table 2: EI results for base year in comparison with the existing regional EI datasets (Gg y⁻¹).**

| Species | Indonesia | | | Thailand | | | Cambodia | | | Other SEA countries[d] | Southern part of China | Total domain |
|---|---|---|---|---|---|---|---|---|---|---|---|---|
| | This study[a] | EDGAR[b] | CGRER[c] | This study[a] | EDGAR[b] | CGRER[c] | This study[a] | EDGAR[b] | CGRER[c] | | | |
| $SO_2$ | 997 | 2,433 | 1,499 | 827 | 721 | 961 | 41 | 42 | 34 | 2,695 | 6,204 | 10,781 |
| $NO_x$ | 3,282 | 2,162 | 1,896 | 701 | 882 | 1,086 | 97 | 92 | 27 | 2,623 | 4,166 | 10,910 |
| CO | 24,169 | 32,246 | 26,703 | 9,095 | 12,553 | 10,815 | 2,877 | 2,453 | 570 | 19,054 | 33,377 | 89,252 |
| NMVOC | 3,840 | 4,528 | 8,225 | 1,120 | 525 | 3,052 | 331 | 18 | 113 | 5,644 | 4,441 | 15,613 |
| $NH_3$ | 1,258 | 1,617 | 1,390 | 469 | 675 | 388 | 110 | 95 | 86 | 1,543 | 2,247 | 5,645 |
| $CH_4$ | 3,950 | 10,300 | 6,443 | 1,053 | 4,541 | 3,567 | 713 | 1,969 | 708 | 13,833 | 14,640 | 34,218 |
| $PM_{10}$ | 2,046 | 3,454 | 1,838 | 782 | 1,196 | 474.9 | 115 | 268 | 68 | 1,763 | 3,644 | 8,458 |
| $PM_{2.5}$ | 1,644 | 2,023 | 1,609 | 607 | 781 | 388.5 | 65 | 201 | 61 | 1,466 | 2,653 | 6,519 |
| BC | 226 | 173 | 229 | 47 | 234 | 72 | 7 | 73 | 7 | 159 | 362 | 821 |
| OC | 674 | 711 | 1,246 | 240 | 73 | 364 | 40 | 13 | 32 | 604 | 643 | 2,245 |
| $CO_2$ | 508,022 | 1,700,450 (514,882) | 587,000 | 260,988 | 235,644 (229,500) | 351,000 | 28,296 | 185,211 (174,300) | 36,000 | 856,225 | 1,406,860 | 3,092,654 |
| $N_2O$ | 180 | 329 | ne | 84 | 71 | ne | 60 | 73 | ne | 271 | 346 | 941 |

Note: ne – not estimated

[a] EI conducted for base year of 2007 using the ABC EIM framework (Shrestha et al., 2013). Detail methodology and results were presented in Permadi et al. (2017).

[b] EDGAR for base year of 2007 (Oliver et al., 2001). Retrieved from http://edgar.jrc.ec.europa.eu/datasets_list.php?v=42. The $CO_2$ emission, excluding forest fire post burn decay and decay of drained peatland, are given in brackets for comparison with our estimates

[c] CGRER for base year of 2006 (Zhang et al., 2009). Retrieved from http://www.cgrer.uiowa.edu/EMISSION_DATA_new/index_16.html. For $NH_3$, $CH_4$, and $CO_2$, emissions were taken from CGRER inventory in 2000 (Streets et al., 2003). Peatland fire for SEA for 2007 was taken from GFED v3. Retrieved from https://daac.ornl.gov/cgi-bin/dsviewer.pl?ds_id=1191.

[d] Other SEA countries include Brunei, Lao PDR, Malaysia, Myanmar, Philippines, Singapore and Vietnam.





**Table 3: Statistical parameters for WRF Model Performance evaluation for two periods**

| Station | Statistical Parameters | | | | | | | | | N |
| | MBE | | | MAGE | | | RMSE | | | |
| | RH (%) | T (ºC) | WS (m s⁻¹) | RH (%) | T (ºC) | WS (m s⁻¹) | RH (%) | T (ºC) | WS (m s⁻¹) | |
| **January –March 2007** | | | | | | | | | | |
| Olongapo-Philippines | 13.5 | 2.7 | 1.3 | 22.7 | 6.2 | 0.18 | 30.2 | 10.5 | 2.7 | 1,861 |
| Davao-Philippines | 29.6 | 10.2 | 2.7 | 38 | 12.8 | 2.9 | 50.4 | 17.4 | 3.4 | 1,250 |
| Don Muang-Thailand | -13 | **-0.4** | 1 | 16 | 2.5 | 1.6 | 18.5 | 3.8 | **2** | 2,148 |
| Trat-Thailand | 38.5 | 14 | 1.6 | 42 | 16.4 | 3.1 | 51 | 20.5 | 3.6 | 996 |
| Pnom Penh-Cambodia | **7.7** | 8.6 | **0.5** | 28 | 10.7 | 2.1 | 35 | 15.4 | 2.6 | 1,513 |
| Jakarta-Indonesia | **-2.6** | 0.6 | 0.7 | 17.8 | 4.6 | 2.1 | 25.8 | 7.5 | 2.6 | 2,036 |
| Kuala Lumpur-Malaysia | **-2.5** | **-0.14** | **0.14** | 6.8 | **1.2** | 1.1 | 10.3 | 2.2 | **1.4** | 2,143 |
| Sarawak-Malaysia | **-1.8** | **-0.13** | **-0.3** | 5.6 | **1.2** | 0.9 | 9.2 | 2.1 | **1.2** | 2,148 |
| **August – October 2007** | | | | | | | | | | |
| Olongapo-Philippines | **5.6** | 6.2 | **0.5** | 16.3 | 6.2 | 1.3 | 26.7 | 8.9 | 3.1 | 1,958 |
| Davao-Philippines | **-0.81** | **-0.13** | **0.2** | 6.4 | 2.2 | 0.7 | 12.8 | 4.8 | **1.3** | 1,262 |
| Don Muang-Thailand | **2.6** | **-0.4** | **-0.1** | 11.1 | 2.1 | 1.4 | 15.4 | 3.5 | 2.2 | 2,139 |
| Trat-Thailand | **0.86** | **-0.1** | 2.1 | 5.4 | **1.3** | 2.7 | 11.6 | 3.8 | 3.3 | 1,017 |
| Pnom Penh-Cambodia | **-6.7** | 0.7 | **0.1** | 10.1 | **1.7** | 1.1 | 14.6 | 3.4 | **1.7** | 1,602 |
| Jakarta-Indonesia | **1.4** | 6.7 | **0.47** | 7.2 | 6.7 | 2.3 | 16.8 | 9.7 | 3.1 | 1,958 |
| Kuala Lumpur-Malaysia | **-5** | **0.3** | **-0.1** | 10.4 | **1.5** | 0.98 | 13.2 | 1.9 | **1.23** | 2,159 |
| Sarawak-Malaysia | **-5.4** | **-0.1** | -0.6 | 8.9 | 3.5 | 1.1 | 11.56 | 4.2 | **1.4** | 2,159 |

Note: bolded values represent satisfactorily model output; Criteria for MBE: WS≤±0.5 m s⁻¹, T≤±0.5ºC, RH≤±10%; Criteria for MAGE: T≤ 2ºC, RH≤ 2 %; criteria for RMSE: WS≤ 2 m s⁻¹, N – number of data points



**Table 4: Statistical parameters for CHIMERE model performance (PM$_{10}$ and BC) evaluation.**

| Parameters and station name | Statistical measures | | |
|---|---|---|---|
| | MBE (µg m$^{-3}$) | MFB (%) | MFE (%) |
| **PM$_{10}$[a]** | | | |
| 1. BMR (average of 10T and 11T)[b] | -17.5 | **-53.3** | **55.7** |
| 2. SUF1 (Surabaya)[c] | -2.6 | **-8.9** | **18** |
| 3. Jerantut, Kuala Lumpur[d] | -13.6 | **-56.3** | **66.5** |
| 4. Petaling Jaya, Kuala Lumpur[e] | -10.3 | **-41.1** | **56.1** |
| **BC[f]** | | | |
| 1. AIT site | -0.12 | **-3.3** | **20.8** |

Note: Criteria from Boylan and Russel (2006). MFB: PM ≤ ±60%, and MFE: PM ≤+75%, bold showed the parameters that satisfy the criteria. No criteria is available for MBE.

[a] Period taken was from January – March and August – October 2007 for all stations (daily average concentrations)

[b] Urban mixed site

[c] Urban mixed site

[d] Background concentration

[e] Urban mixed site

[f] Period taken was from March to December 2007 (daily average concentrations).





**Figure captions**

**Figure 1:** Gridded (0.25º x 0.25º) annual emissions for the selected pollutants over the SEA domain: a) BC and b) CO
**Figure 2:** Comparison of modeled and observed 24h $PM_{10}$ in Kuala Lumpur, Malaysia (1 station), Surabaya, Indonesia (1 station) and Bangkok, Thailand (3 stations). Note that the stations included in the comparison are those

10 located within the cell.
**Figure 3:** Scatter plots of modeled vs. observed 24h $PM_{2.5}$ at four AIRPET sites, 2007
**Figure 4:** Time series comparison and scatter plot of modeled vs. observed 24h EC in AIT site, 2007
**Figure 5:** Comparison of 24h simulated and observed BC at four AIRPET sites in SEA domain, 2007
**Figure 6:** Spatial distribution of monthly average $PM_{10}$, $PM_{2.5}$ and BC in the selected months, 2007

15 **Figure 7:** Spatial distribution of monthly modeled AOD as compared to the MODIS Terra AOD for the selected months, 2007
**Figure 8:** Monthly average of simulated vs. observed AOD at 10 AERONET stations, 2007




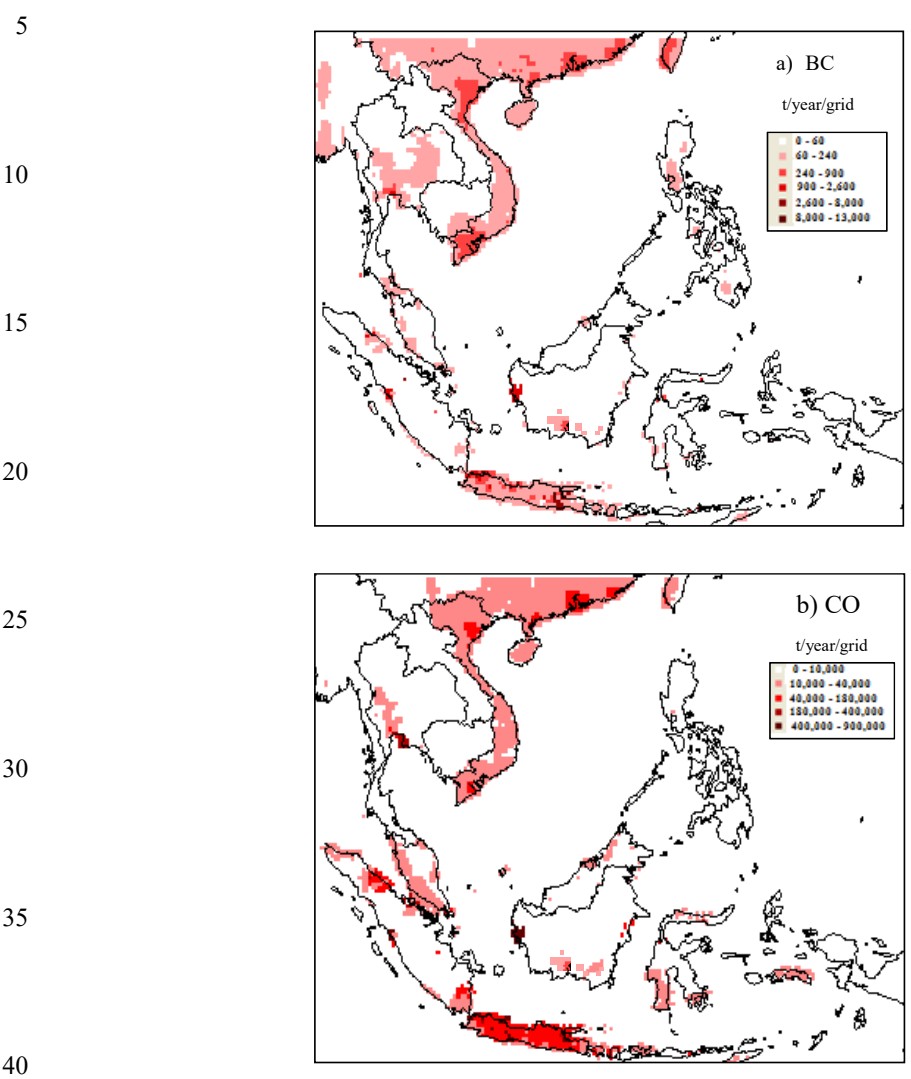

**Figure 1: Gridded (0.25º x 0.25º) annual emissions for the selected pollutants over the SEA domain: a) BC and b) CO.**





**Figure 2: Comparison of modeled and observed 24h PM$_{10}$ in Kuala Lumpur, Malaysia (1 station), Surabaya, Indonesia (1 station) and Bangkok, Thailand (3 stations). Note that the stations included in the comparison are those located within the cell.**



**Figure 3: Scatter plots of modeled vs. observed 24h PM$_{2.5}$ at four AIRPET sites, 2007.**





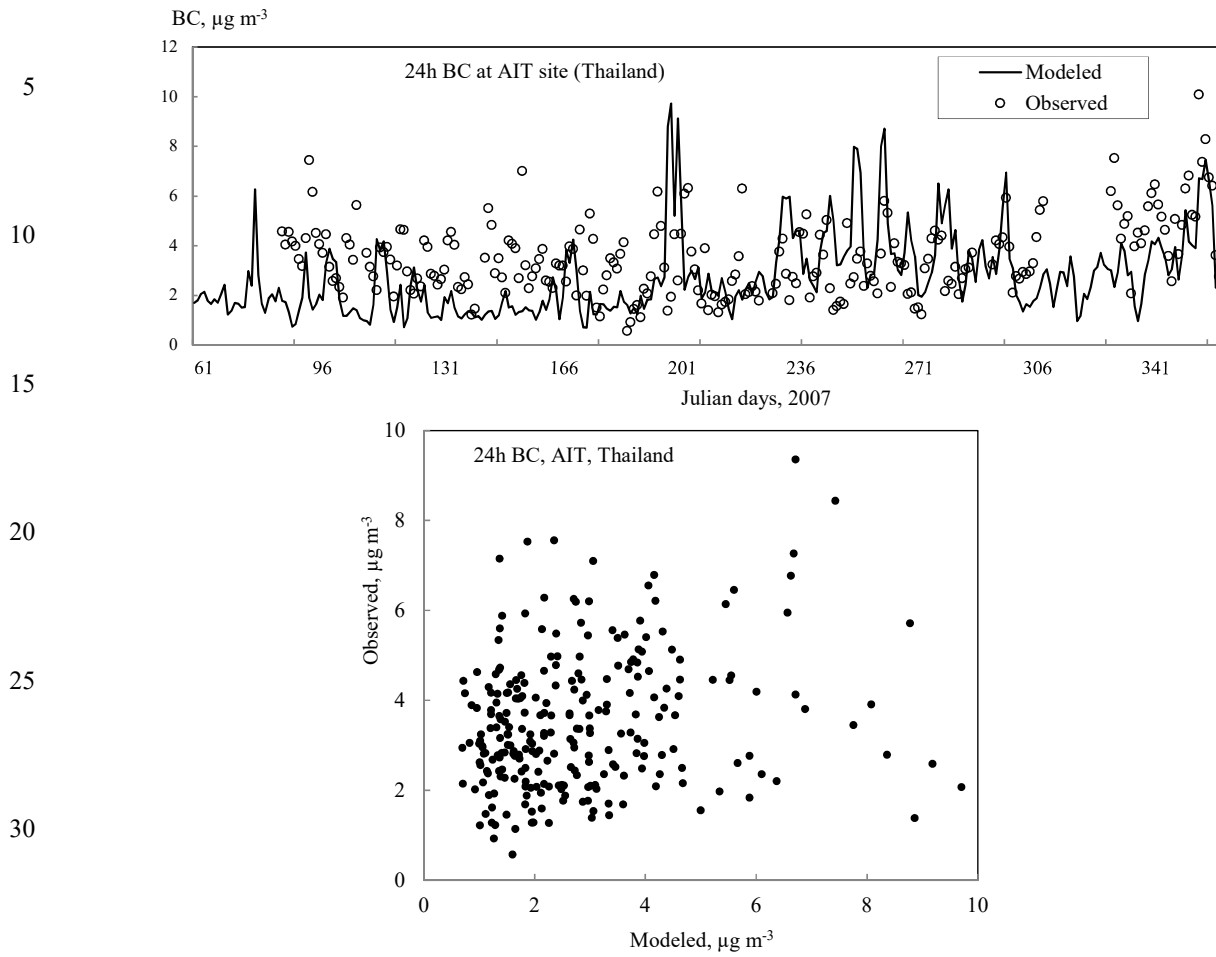

**Figure 4: Time series comparison and scatter plot of modeled vs. observed 24h EC in AIT site, 2007.**





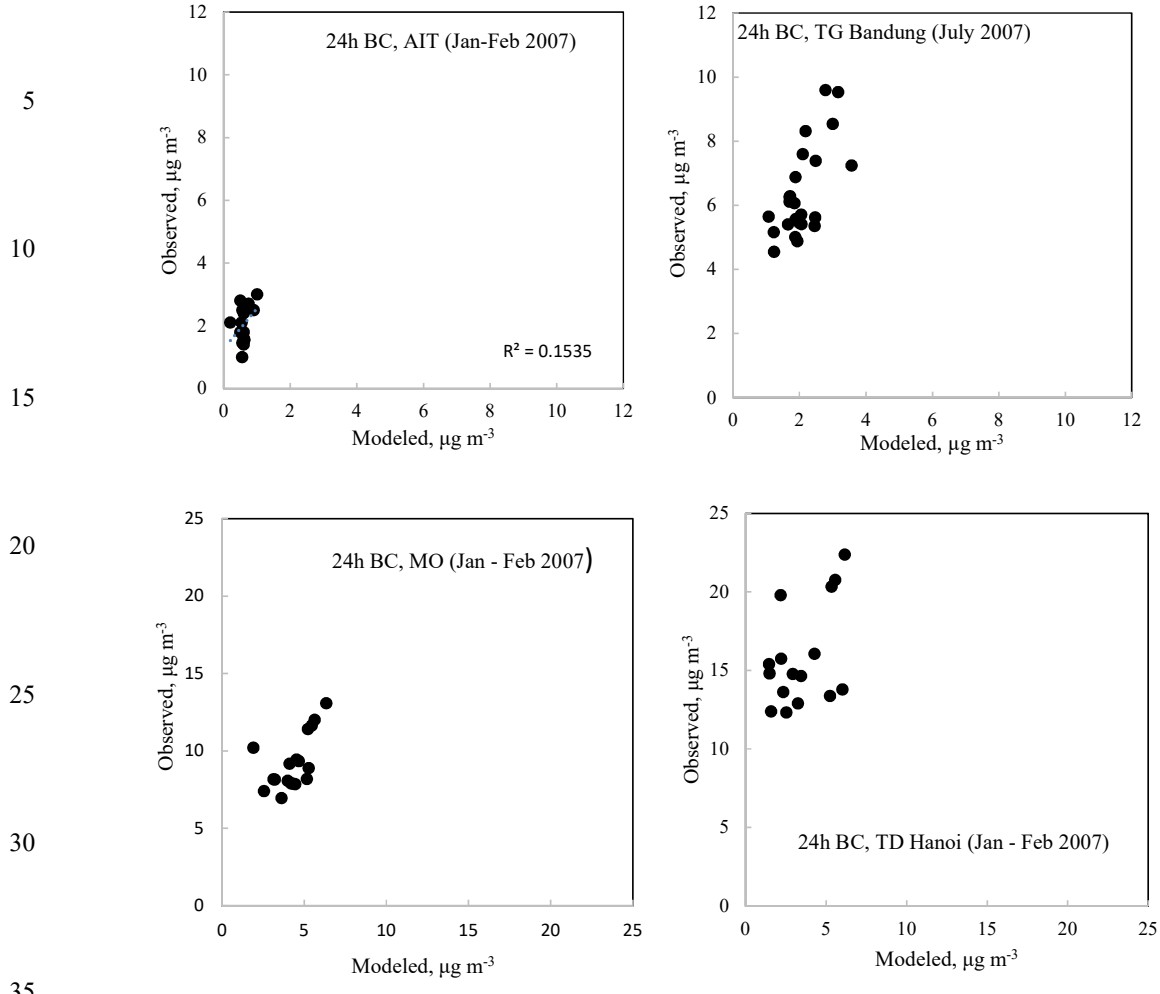

**Figure 5: Comparison of 24h simulated and observed BC at four AIRPET sites in SEA domain, 2007.**





**Figure 6: Spatial distribution of monthly average PM₁₀, PM₂.₅ and BC in the selected months, 2007.**





**Figure 7: Spatial distribution of monthly modeled AOD as compared to the MODIS Terra AOD for the selected months, 2007.**





**Figure 8: Monthly average of simulated vs. observed AOD at 10 AERONET stations, 2007.**