# Peer review of "Integrated emission inventory and modeling to assess distribution of particulate matter mass and black carbon composition in Southeast Asia"

_Atmospheric Chemistry and Physics, 2017_

## Referee Comment (RC1) · Anonymous Referee #1 · 18 Jun 2017

The authors developed an anthropogenic emission inventory based on various published emission data including their own. With the emission inventory, the WRF model simulation has been conducted for the year 2007 and the results are compared to in situ observations of PM10, PM2.5 and BC concentration from various stations in Southeast Asia. Methodology and analysis are sound in general, but the authors mostly focused on comparing above-mentioned variables while meteorology components are almost completely excluded in the discussion on simulated biases. Including the analysis of the simulated metrological fields will be beneficial to understand the similarity and dif-

ference between observation and simulation. I recommend that the manuscript require major revision prior to publication.

Specific Comments:

1) This is the first one of two part papers. The title doesn't seem to represent this manuscript very well. It should be just two papers rather than two parts.

2) Page 5, lines 15-18: Surface concentration and PM may not be affected by this, but it may change total column concentration and AOD. What's the reason for this limitation? Is nudging filled the layers above 500hPa?

3) Page 6, line 31: Is this AOD for surface to 500hPa? Does this mean no long-range transported aerosol are considered in this study?

4) Page 7, line 10, "interannual variation in forest fire": Does the emission inventory used in this study include fire emission in 2007 (See also line 25 on page 3)?

5) Lines 27-32 on Page 8 & lines 25-26 on Page 9: It would be a good idea to show model simulation of winds (probably at 850hPa) and rainfall to discuss a possible impact of transport and scavenging effects to examine the discrepancy between model and observation.

6) Line 5 on Page 10, "caused by the limited monitoring data availability": Not clear. Not enough emission data? Could you elaborate this?

7) Lines 4-20 on page 12: PM2.5/PM10 ratio is used as a proxy to show the dominance of local sources. However, if the model top layer is 500hPa (Page 5, lines 15-18), aren't remote-origin aerosols suppressed in the model?

8) Lines 23-24 on page 12: All observations are from big cities. Can this ratio still provide any additional information?

9) Figure 6 & lines 30-34 on Page 13: All plumes seem to converge in the South China Sea. What's the role of meteorology, rain and winds?

10) Lines 1-2 on page 14: It is not clear. Please elaborate.

11) Figure 7: Same color scheme should be used for better comparision.

12) Lines 7-8 on page 15, "better seasonal variability": Seasonal cycle in Phimai and Mukdahan are not simulated.

13) Figure S5: BC AOD in August is same as BC AOD in November. Please check.

---

## Referee Comment (RC2) · Anonymous Referee #2 · 25 Aug 2017

There is no new method or new findings in this paper. However, the model domain, the SEA area, which the authors thought necessary to evaluate, is the important point of this paper. Despite many restrictions, such as few observation data and the target year of 2007 seeming rather old, the results are reasonable.

There is black carbon (BC) in the title, however, I cannot recognize the necessity of explicitly mentioning BC. It should be connected with contents of Part 2, but it may be written as PM?

In addition, as the title is "emission inventory", especially Indonesia, Thailand, and Cambodia seem to be using detailed EI in each region, please write those contents more carefully.
The model domain contains a lot of sea area, so please mention emissions from ships.

< Specific Comments >
p.2 L9
Since the annotation of PM$_{2.5}$ is carefully written, I think that it is better to write PM $_{10}$ as well.

p.3 L16
 "The emissions from major anthropogenic sources (except for biomass open burning) in Indonesia, Thailand and Cambodia"
Please describe that part in detail, especially, temporal and spatial distribution. Table 1 contains only activity data used for EI calculation. It does not match the description in the text.

p.3 L23
For CROB, Kanabkaew and Kim Oanh (2011) and Permadi and Kim Oanh (2013) are extremely fine in both spatial and temporal resolutions to apply CTMs. I think that it should be described in detail. Furthermore, if it is possible, please indicate the difference between those data and GFEDv3.
Do biomass burning emissions of Cambodia depend on GFEDv3?

p.5 18L
I think that the boundary condition is a bit old. Were there problems to adapt 1998-2002

year conditions to the 2007 year? It is better to apply the result of global chemistry modeling such as MOZART or CHASER.

p.6 17L and p.10 13L etc.,

I think that AIT refers to the Asian Institute of Technology, but please describe it clearly.

p.7 26L~

It is impossible for Cambodia, $CO_2$ of EDGAR emission to be a reasonable agreement. For Thailand and Cambodia, BC of EDGAR emission is high but Indonesia is not so high. I think that it can be safely said that CGRER of OC is certainly higher for Indonesia, however, for Thailand, OC of CGRER is not so high, rather OC of EDGER is low.

p.7 31L

Which of the legends of Figure S2, SI refers to commercial combustion?

p.8 6L

It is Table 2 instead of Table 3.

p.8 11L

Why are the authors showing CO (Carbon monoxide) in many chemical species in Fig.1?

p.8 "3.2 WRF model results and evaluation" and/or Table 3

Please note the description of the statistics measures are listed in Table S1, SI.

p.9

There is "1 station in Kuala Lumpur, 2 stations in Bangkok, and 1 station in Surabaya" in the text, but in Figure 2,

      1 station in Kuala Lumpur

      1 station in Surabaya

      3 stations in Bangkok

are described. Is Bangkok 3 stations or 2 stations?

Also, in Table 4, not the notation, it is difficult to understand the meaning of BMR and SUF 1, so please specify the place name as shown in the main text and Figure 2.

It might be better to list measuring station names. Since there is no explanation of the meanings of 10T and 11T, are there any relationships with 13T and 20T of Ground measurement of aerosols, 4, Urban air quality monitoring network in Bangkok, Thailand in Table S2, SI? Or if those station names are not important I think that they might be omitted.

p.11 8L

Please explain scientifically and carefully the difference between EC and BC.

p.12 9L

As authors wrote in the text "$PM_{2.5}$ are more uniformly distributed in an urban area than the coarser particles", the value of $PM_{2.5}/PM_{10}$ be influenced not by local $PM_{2.5}$ emissions, but rather by local $PM_{10}$ emissions?

p.13 14L

Maximum monthly average of BC in Aug. and Nov. are different from the value of Table S3, SI. Please check it.

p.15 11L

In the text, "The seasonal variation in the emission input file would need to be further refined to improve the situation". However, the data of GFED and the biomass burning data which authors applied are changing at least monthly. Does it refer other anthropogenic emissions rather than biomass burning?

< Tables and Figures >

Figure S1, SI

Characters in the legend of Figure S1, SI are collapsed and cannot be read.

Figure S2, SI

$PM_{2.5}$ and $PM_{10}$ on the vertical axis of the Figure S2, SI are written as $PM_2$ and $PM_1$, respectively.

The sizes (heights) of the graphs of Indonesia, Cambodia and Thailand are different. The height of Cambodia's graph is high and that of Thailand is low.

---

## Author Comment (AC1) · 10 Oct 2017

REVIEWER 1# The authors developed an anthropogenic emission inventory based on various published emission data including their own. With the emission inventory, the WRF model simulation has been conducted for the year 2007 and the results are compared to in-situ observations of PM10, PM2.5 and BC concentration from various stations in Southeast Asia. Methodology and analysis are sound in general, but the authors mostly focused on comparing above-mentioned variables while meteorology components are almost completely excluded in the discussion on simulated biases.

[Figure]

Including the analysis of the simulated metrological fields will be beneficial to understand the similarity and difference between observation and simulation.

Response: Thank you for your useful comments and we appreciate the suggestion to include the analysis of the simulated meteorological fields. In the revised version, we presented more detail results of WRF evaluation and the simulated results of PM and BC have been discussed considering the meteorology simulation.

Specifically for WRF results, Section 3.2.2 Synoptic scale model evaluation was added to the previously presented 3.2 (now 3.2.2) "Statistical performance evaluation of WRF". We included analyses of simulated synoptic pressure, upper wind (at 850 hPa) and monthly precipitation in comparison with the observations. Namely, the simulated synoptic surface pressure and upper wind fields were compared with the weather charts provided by the Thailand Meteorological Department (TMD), as detailed in Figure S4 and Figure S5. Simulated monthly precipitations for two selected months (i.e. August and October) were compared with the Tropical Rainfall Measuring Mission (TRMM-3B43) dataset in the newly added Figure 2.

In the discussion of PM model results, the effect of meteorological model results were referred in addition to the other factors, such as the grid averaging (30x30 km2) effects of the model results as compared to point by point observation, and the uncertainty in the emission inventory results. Detailed responses to the comments are included below.

I recommend that the manuscript require major revision prior to publication. Specific Comments: 1) This is the first one of two part papers. The title doesn't seem to represent this manuscript very well. It should be just two papers rather than two parts.

Response:

Thank you for the suggestion. The title has been revised and we do not use part 1 and part 2 in the revised versions. This paper is now titled "Integrated emission

inventory and modeling to assess distribution of particulate matter mass and black carbon composition in Southeast Asia". We also revised the abstract accordingly.

2) Page 5, lines15-18: Surface concentration and PM may not be affected by this, but it may change total column concentration and AOD. What's the reason for this limitation? Is nudging filled the layers above 500 hPa?

Response:

We used the previous version of the model (CHIMERE 2008c) with a simplified photolysis rate calculation (Madronich et al., 1998) that assumed the model domain is below the cloud, hence putting a constraint on model top to be maximum at 500 hPa. It is true that with the current vertical set-up the long range transport (LRT) that takes place in the free tropospheric layer may not be captured hence it may cause bias in the total column AOD calculation when LRT is substantial.

However, we also checked the PBL and the maximum value throughout simulation period in 2007 was 3,900 m (Figure S6), i.e. below the upper level of the domain (5,500 km), hence suggesting that the mixing down effects of the LRT pollution from above PBL may be partly included. The nudging filled the layers only until the PBL layer in the WRF model set-up. We recommend for the future studies to use latest version of CHIMERE, for example version 2014, with updated radiative transfer model that can compute photolysis reaction rates for extending the model domain vertically beyond 500 hPa. In SEA, the aerosol extinction coefficient in the transboundary haze affected sites observed by the level 2 NASA Cloud Aerosol Lidar with Orthogonal Polarization (CALIOP) was not significant above the vertical height of 5 km (Campbell et al., 2013). However, we realized potential bias of the current vertical layer set-up in relation to the LRT contribution and in the revised version we emphasized on this limitation and potential bias introduced by the current set-up in lines 2-8, page 6.

3) Page 6, line 31: Is this AOD for surface to 500hPa? Does this mean no long-range transported aerosol are considered in this study?

[Figure]

Response:

Yes. AOD was calculated from the surface to the model top layer of 500 hPa. The restriction of extending the model vertical layer beyond of 500 hPa was explained in the response to the comment no. 2. It is true that LRT may be limited and only the transport within the PBL was actually considered in this simulation. Discussion was added in the lines 30-32, page 6 and also the reason was added when AOD was underestimated in comparison with the satellite AOD in lines 26-28, page 16.

4) Page 7, line 10, "interannual variation in forest fire": Does the emission inventory used in this study include fire emission in 2007 (See also line 25 on page 3)?

Response:

Yes. The emission inventory included forest fire emission for the base year of 2007, and we revised the sentence in line 30, page 3 to make it clear on the base year of fire emissions. We deleted the "interannual variation in forest fire" in the new revised sentence in lines 6-8, page 8.

5) Lines 27-32 on Page 8 & lines 25-26 on Page 9: It would be a good idea to show model simulation of winds (probably at 850hPa) and rainfall to discuss a possible impact of transport and scavenging effects to examine the discrepancy between model and observation.

Response:

Thank you for the useful suggestion. We have added comparison between simulated wind fields at the level of 850 hPa with the synoptic upper wind from the TMD in Figure S5 and a discussion was added in lines 11-14, page 10.

Figure 2 was also added to present comparison of modeled monthly precipitations with the TRMM-3B43 dataset. Model results could capture the upper synoptic wind at 850 hPa. The precipitation patterns were well reproduced but the domain maximum monthly values were underestimated. A discussion was added in lines 14-17, page

10. We further referred to the meteorology simulation in the discussion of the model performance of PM10 (lines 25-28, page 11).

6) Line 5 on Page 10, "caused by the limited monitoring data availability": Not clear. Not enough emission data? Could you elaborate this?

Response:

Thank you. In addition to the simulated meteorological parameters and EI, we also highlighted the availability of the hourly dataset of PM10 from the government network of the automatic monitoring stations which was not sufficient to conduct a comprehensive comparison. There were missing hourly missing data in this particular station especially for the period of January – March 2007, hence the episodic values might not be recorded. The elaboration is in lines 28-29, page 11.

7) Lines 4-20 on page12: PM2.5/PM10 ratio is used as a proxy to show the dominance of local sources. However, if the model top layer is 500hPa (Page5, lines15-18), aren't remote-origin aerosols suppressed in the model?

Response:

Thank you for your comment and as responded before we agreed that LRT above PBL may not be captured because of the current vertical model set-up. This would affect more PM2.5 and less the coarse fraction (PM10-2.5). Nevertheless, several local combustion sources also contribute considerable amounts of PM2.5, such as on-road traffic and residential combustion, and this affect the PM2.5 hence the ratio. The discussion of PM2.5/PM10 ratios (for model 1st layer) in Section 3.3 as a part of the model result evaluation may explain the discrepancy between modeled and observed PM concentrations using the available monitoring data. The revision was made in lines 4-5, page 15.

8) Lines 23-24 on page 12: All observations are from big cities. Can this ratio still provide any additional information?

Response:

The lack of data for the areas outside the cities is an issue remained. Generally, we expect that PM2.5 mass may be more uniform in an urban area, for example measurements conducted in several mountain areas in Asia showed high PM2.5 concentrations which were mainly due to the regional transport (Hang and Kim Oanh, 2014; Co et al., 2014) or local combustion sources (e.g. residential cooking, biomass OB) such as found in China (Liu et al., 2017). However, the BC fraction of PM may vary a lot with much lower values in remote sites. However, lack of the data would prevent from a more in-depth analysis and we do not make any recommendation on the ratios. We however, added sentences referring to this issue in lines 34-37, page 14 and lines 1-2, page 15.

9) Figure 6 & lines 30-34 on Page 13: All plumes seem to converge in the South China Sea. What's the role of meteorology, rain and winds?

Response:

The plumes of PM10 and PM2.5 were converged in the South China Sea in January and November when the NE monsoon prevalent that brought PM pollution from the Southern part of China mainland to the South China Sea. Figure S8 was added to give highlight how the NE wind governed the convergence pattern of PM in the South China Sea in the selected days in both months. WRF result showed no rainfall over the South China Sea during the particular period that may also contribute to the converged PM levels. Discussion was added in the revised manuscript in lines 1-4, page 16.

10) Lines 1-2 on page 14: It is not clear. Please elaborate.

Response:

Thank you for the comment. The sentence was revised in lines 34-35, page 15.

11) Figure 7: Same color scheme should be used for better comparison.

Response:

Thank you for your suggestion. The figures were revised accordingly (Figure 8). 12) Lines 7-8 on page 15, "better seasonal variability": Seasonal cycle in Phimai and Mukdahan are not simulated.

Response:

Agree, we removed the "better seasonal variability" and the sentence was revised to highlight different performance in different stations in lines 10-13, page 17.

13) Figure S5: BC AOD in August is same as BC AOD in November. Please check.

Response:

Thank you for the correction. The figure for BC AOD in November was corrected (Figure S9, SI).

  REVIEWER 2#

There is no new method or new findings in this paper. However, the model domain, the SEA area, which the authors thought necessary to evaluate, is the important point of this paper. Despite many restrictions, such as few observation data and the target year of 2007 seeming rather old, the results are reasonable. There is black carbon (BC) in the title, however, I cannot recognize the necessity of explicitly mentioning BC. It should be connected with contents of Part 2, but it may be written as PM?

Response:

Agree. Title was changed to "Integrated emission inventory and modeling to assess distribution of particulate matter mass and black carbon composition in Southeast Asia". We still mention BC because it is our primary concern to link with the co-benefit study especially on the climate forcing. We did not put part 1 and part 2 anymore but rather as two separate papers. We also recognize that a more recent base year would be good but as our purpose was the model evaluation and we have more PM and BC

data in 2007 as compared to other years hence 2007 was selected in the first place.

In addition, as the title is "emission inventory", especially Indonesia, Thailand, and Cambodia seem to be using detailed EI in each region, please write those contents more carefully. The model domain contains a lot of sea area, so please mention emissions from ships.

Response:

Thank you for your suggestion. We have provided a dedicated Section 3.1 to discuss the emission inventory development for the three target countries including the comparison of the results with the existing inventories. We referred to the detail framework of emission inventory presented in our previous papers such as Permadi et al. (2017), Permadi and Kim Oanh (2013), and Kanabkaew and Kim Oanh (2011) with regards to the activity data, selection of the emission factors, temporal and spatial distribution. We did not include the emission from the international shipping in this simulation due to the lack of the data. However the "inland waterway" source was included in the inventories. A sentence was added to elaborate this point in the revised version of the manuscript, lines 6-7, page 4. Uncertainty associated with the missing source like international shipping was added in line 4, page 17.

< Specific Comments > p.2 L9 Since the annotation of PM2.5 is carefully written, I think that it is better to write PM10 as well.

Response:

Thank you for your suggestion. Detail annotation was added for PM10 in line 9, page 2. The new sentence is: "Being components of PM, e.g. PM2.5 and PM10 (PM with diameter less than 10 micron), black carbon (BC) and organic carbon (OC), have been monitored in some Asian cities and the results, although fragmented, showed considerably high levels (Kondo et al., 2009; Kim Oanh et al., 2006; Hopke et al., 2008)."

p.3 L16 "The emissions from major anthropogenic sources (except for biomass open

burning) in Indonesia, Thailand and Cambodia" Please describe that part in detail, especially, temporal and spatial distribution. Table 1 contains only activity data used for EI calculation. It does not match the description in the text.

Response:

The sentence in line 16 page 3 was revised to be as follows: "The emissions from major anthropogenic sources in Indonesia, Thailand and Cambodia were developed following the EI framework given in the Atmospheric Brown Cloud Emission Inventory Manual (ABC EIM) (Shrestha et al., 2013), using the activity data summarized in Table 1". Additional explanation on the spatial distribution of emissions was added in lines 17-21, page 3 and an explanation on the temporal variation of emissions was added in lines 13-22, page 4.

p.3 L23 For CROB, Kanabkaew and Kim Oanh (2011) and Permadi and Kim Oanh (2013) are extremely fine in both spatial and temporal resolutions to apply CTMs. I think that it should be described in detail. Furthermore, if it is possible, please indicate the difference between those data and GFEDv3. Do biomass burning emissions of Cambodia depend on GFEDv3?

Response:

Thank you for the comment and query. We highlighted in the previous response to address the temporal and spatial distribution and also to referred to the relevant publications. The difference between those data and GFEDv3 was added in lines 32-35, page 3. Sentence was added in line 31-32, page 3, to state that the CROB emission was included in our EI for Cambodia however forest fire emission for the country was obtained from the GFEDv3 database.

p.5 18L I think that the boundary condition is a bit old. Were there problems to adapt 1998-2002 year conditions to the 2007 year? It is better to apply the result of global chemistry modeling such as MOZART or CHASER.

Response:

We used the boundary conditions consisting of the monthly mean concentrations of species obtained from the global simulation of LMDz-INCA for the period of 1998-2002. This dataset was provided in the CHIMERE v2008c as suggested in the user guide and it has been widely used by the users of this version of CHIMERE. We however recognize the potential effects of the aged boundary conditions and compare these with the monthly mean concentrations of 2007. The ratios of concentrations of respective species between 2 datasets (C2007 / C1998-2002) for aerosol and PM precursor gases (i.e. BC, OC, $NO_2$, CO, $SO_2$, $SO_4$, $C_2H_4$, $CH_3CHO$, and $NH_3$) ranged from 0.98 – 1.23. This implies that basically the two datasets were almost similar. The impacts of the aged boundary conditions on the simulation are expected but with a small magnitude. A discussion was added in section 2.3 lines 13-17, page 6.

p.6 17L and p.10 13L etc., I think that AIT refers to the Asian Institute of Technology, but please describe it clearly.

Response:

Thank you very much for the suggestion. Full explanation of AIT is now added in line 23, page 4 for the Asian Institute of Technology.

p.7 26L~ It is impossible for Cambodia, CO2 of EDGAR emission to be a reasonable agreement. For Thailand and Cambodia, BC of EDGAR emission is high but Indonesia is not so high. I think that it can be safely said that CGRER of OC is certainly higher for Indonesia, however, for Thailand, OC of CGRER is not so high, rather OC of EDGER is low.

Response:

Thank you. We are sorry to misplace BC and OC data quoted from EDGAR for Thailand and Cambodia. We revised the values in Table 2 accordingly. For Thailand, EDGAR BC and OC values should be 73 and 234 Gg yr-1, respectively. For Cambodia, EDGAR BC and OC values should be 13 and 73 Gg yr-1. The values of BC and OC for all 3 databases are more consistent and comparable after the revision. The discussion in the text (lines 28-29, page 8) was revised accordingly. It is now also highlighted that CO2 of EDGAR was far higher than the study and CGRER in lines 26-28, page 8.

p.7 31L Which of the legends of Figure S2, SI refers to commercial combustion?

Response:

Commercial combustion was accounted under the Residential. Figure S2, SI legend was revised, the light green belongs to "Residential and commercial combustion". Thank you for the correction.

p.8 6L It is Table 2 instead of Table 3.

Response:

Thank you for the correction. It was revised accordingly to refer to Table 2 in line 6 page 9.

p.8 11L Why are the authors showing CO (Carbon monoxide) in many chemical species in Fig.1?

Response:

We presented BC and CO in Fig. 1 to highlight the primary aerosol and trace gas emissions. Both species are product of incomplete combustion especially from the contained combustion and biomass open burning.

p.8 "3.2 WRF model results and evaluation" and/or Table 3 Please note the description of the statistics measures are listed in Table S1, SI.

Response:

We have now stated in Section 2.5 Model evaluation to refer readers to the Table S1, SI

for the definition of the statistical measures used in the model performance evaluation (line 23-24, page 7). A footnote was added in Table 3 to refer to Table S1, SI for the details.

p.9 There is "1 station in Kuala Lumpur, 2 stations in Bangkok, and 1 station in Surabaya" in the text, but in Figure 2, 1 station in Kuala Lumpur 1 station in Surabaya 3 stations in Bangkok are described. Is Bangkok 3 stations or 2 stations? Also, in Table 4, not the notation, it is difficult to understand the meaning of BMR and SUF 1, so please specify the place name as shown in the main text and Figure 2. It might be better to list measuring station names. Since there is no explanation of the meanings of 10T and 11T, are there any relationships with 13T and 20T of Ground measurement of aerosols, 4, Urban air quality monitoring network in Bangkok, Thailand in Table S2, SI? Or if those station names are not important I think that they might be omitted.

Response:

Thank you for the correction. Figure 2 was revised to highlight only 2 stations in Bangkok (10T and 11T). Accordingly, Table S2, SI was also revised to explain more detail of the station 10T and 11T in Bangkok. We keep the name of 10T, 11T and SUF1 as they are commonly used in the respective country with an explanation in Table S2, SI.

p.11 8L Please explain scientifically and carefully the difference between EC and BC.

Response:

The sentence was improved with the addition on operational definition of BC and EC, i.e. the different measurement methods; line 4 page 13.

p.12 9L As authors wrote in the text "PM2.5 are more uniformly distributed in an urban area than the coarser particles", the value of PM2.5/PM10 be influenced not by local PM2.5 emissions, but rather by local PM10 emissions?

Response:

[Figure]

We have revised this part accordingly that is also to reflect the comment from Reviewer 1. Accordingly, the potential contribution of LRT was mentioned in the text (lines 1-5, page 14) that may contribute more to PM2.5 than the coarse fraction of PM10-2.5. We mentioned that variations in the PM2.5/PM10 ratios, contributions of various sources of the coarse particles (PM10-2.5), such as road dust and construction dust, should be further analyzed. It is noted that that the ratios used to compare with the model simulated values were all derived from the observations made in large cities in SEA. Lack of the observation data in rural areas and remote sites presents an obstacle for more in depth analysis of the model performance. A discussion was added in the text, lines 15-19, page 14.

p.13 14L Maximum monthly average of BC in Aug. and Nov. are different from the value of Table S3, SI. Please check it.

Response:

Thank you for your correction, the values in the text were misquoted. The sentence has been revised using the correct data presented in Table S3, SI (line 21, page 15).

p.15 11L In the text, "The seasonal variation in the emission input file would need to be further refined to improve the situation". However, the data of GFED and the biomass burning data which authors applied are changing at least monthly. Does it refer other anthropogenic emissions rather than biomass burning?

Response:

Thank you for the query. We used proxies for the monthly variations of emissions of several important sources (i.e. transport, industry, residential, power plant, etc.) because of the lack of the monthly activity data. Only CROB and forest fire emissions have been developed using monthly based activity data directly. The sentence was revised accordingly to explain better in lines 24-26, page 17.

< Tables and Figures > Figure S1, SI Characters in the legend of Figure S1, SI are

collapsed and cannot be read.

Response:

Thank you for the correction. We corrected the Figure S1, SI legend accordingly.

Figure S2, SI PM2.5 and PM10 on the vertical axis of the Figure S2, SI are written as PM2 and PM1, respectively. The sizes (heights) of the graphs of Indonesia, Cambodia and Thailand are different. The height of Cambodia's graph is high and that of Thailand is low.

Response:

Thank you for your useful suggestion. Figure S2, SI was revised accordingly to improve the vertical axis and the height of the graphs as well as the vertical axis legend. The copying left out some of the letters, and we improved it in this version.

Please also note the supplement to this comment:
https://www.atmos-chem-phys-discuss.net/acp-2017-315/acp-2017-315-AC1-supplement.pdf

[revised manuscript text omitted]

a) Modeled surface pressure in 1st January 2007, 07:00 LST

b) Synoptic weather chart in 1st January 2007, 07:00 LST

c) Modeled surface pressure in 8 September 2007, 07:00 LST

d) Synoptic weather chart in 8 September 2007, 07:00 LST

**Figure S4 Comparison of modeled surface pressure and synoptic weather charts**

a) Modeled upper wind field at 850 hPa on 8 October 2007, 07:00 LST

b) Synoptic upper wind field at 850 hPa on 8 October 2007, 07:00 LST

m s⁻¹

d) Modeled upper wind field at 850 hPa on 7 November 2007, 07:00 LST

c) Synoptic upper wind field at 850 hPa on 7 November 2007, 07:00 LST

**Figure S5 Modeled upper wind fields and synoptic wind at 850 hPa on selected days**
**Note: A – Anticyclone (high pressure system)**

**Figure S6 Domain maximum hourly values of simulated PBL for the different months**

**Figure S7: Simulated annual average concentration of 2007: a) PM$_{10}$, b) PM$_{2.5}$ and c) BC over the SEA domain.**

a) Simulated hourly PM$_{10}$, 22 January 2007, 07:00 LST

PM10 Concentration (ug/m3)

2.1E-02    1.4E+01    2.8E+01    4.1E+01    5.5E+01    6.9E+0

b) Simulated hourly PM$_{10}$, 20 November 2007, 07:00 LST

d) Simulated hourly wind field (10 m), 22 January 2007, 07:00 LST

Sqrt[(U10)² + (V10)²]  (m s-1)

6.6E-03    3.0E+00    6.1E+00    9.1E+00    1.2E+01    1.5E+01    1

c) Simulated hourly wind field (10 m), 20 November 2007, 07:00 LST

e) Simulated hourly precipitation, 22 January 2007, 07:00 LST

(mm)

5.3E-09    3.3E+02    6.5E+02    9.8E+02    1.3E+03    1.6E+03

f) Simulated hourly precipitation, 20 November 2007, 07:00 LST

**Figure S8: Typical simulated maximum hourly PM$_{10}$ concentration and wind field at 10 m in 22 January and 20 November, 07:00 LST.**

**Figure S9: Monthly average of simulated BC AOD for the selected months.**

---

## Referee Report (RR1)

I would like to thank the authors for their overall revised content. I thought that they carefully answered the points from each reviewer. I would like to note some points that I am concerned about, including some points I did not notice in the previous review.

<Abstract>

The authors commented that this paper was intended to be composed of Part 1 and Part 2 but would handle it as two separate papers. However, Permadi et al.'s title on the reference has not been modified, and it remains as Part 2. Although it is not directly related to this paper, I think that it is better to modify it.

<text>

p.3 L7

"The results of this Paper 1 are used in the follow up study which investigated the potential co-benefits of various emission reduction measures implemented in Indonesia and Thailand on air quality improvement, number of premature death reduction and climate forcing mitigation in 2030 (Permadi et al., 2017a)."
It remains a composition of Part 1 and Part 2.

p.3 16L

There is a description of spatial distribution, but the description of OB is followed in this part. I think that it would be better to add a description on spatial distribution of OB or to put the description of spatial distribution after the OB description.

p.3 L32 and p.4 L1

Explanation of GFED3 abbreviation is coming after the first appearance. An abbreviation should be added after the first case the term is used.

p.4 L5

The description of the ship is in the paragraph of CROB emission.

p.4 L10

The listed URL (http: //glcf.umi aces.umd.edu) did not reach the GFCL site. Please check it.

p.4 L11

Is the expression "NOx emissions from natural vegetation." correct? "NO emissions from soil"?

p.6 L13

The authors compared boundary concentrations of 2007 and 1998-2002, and the

difference was described as being in the range of 0.98 - 1.23. However, the authors' consideration about this value is not shown in the text. As in the "authors' response to reviewer", it needs a comment such as "This implies that basically the two datasets were almost similar. The impacts of the aged boundary conditions on the simulation are expected but with a small magnitude".

p.10 L14
Why do the authors show examples in October and November to compare the wind situation of the upper layer with observations?

p.10 L20
Please write the approximate altitude of 500 hPa as shown on p.10 L12.

p.13 L4
I think that it is better to explain this part "EC was measured using thermal optical method while BC was measured using light absorption method." at the measurement method (p.12 L29 or L30). In addition, please specify that there is no problem about comparison of EC and BC directly.

p.15 L10
Is the value shown in Table S3 the average value of the entire model domain? Please write it clearly.

p.17 L13
The position of 10 AERONET stations can also be added in Figure S1. Or I think that it is good to put them in Figure 8.

<Reference>
p.24 L5
Is 4 of "UNEP-C4." a superscript?

<Figures>
Figure 1
The letter CO in b) has disappeared.

Figure 6
I am sorry I could not point it out in the last review, but I think that "PBCAR" in the legend should be "BC".

Figure 8

Legend numbers are too small to read.

Figure S4
The calculation result and the weather chart are difficult to compare. Since the calculation result is drawn with surface pressure, the information of the terrain in the figure is highlighted and it is difficult to compare the pressure pattern. Please draw to sea level pressure uniformly.

Figure S7
As in Figure 7, it is better to write "PM 2.5 January" in the figure.

---

## Author Response (AR2)

**Nov. 25, 2017**

Dear editor,

Please find the revised version of our MS entitled "Integrated emission inventory and modeling to assess distribution of particulate matter mass and black carbon composition in Southeast Asia". We have revised our MS taking into account useful suggestions from both reviewers and revisions are marked with blue color. The major change is the inclusion of comparison of surface pressure (produced by WRF) with the ERA interim reanalysis data presented in Figure S4 and S5 and also additional discussions on the effects of precipitation on PM and AOD levels.

Thank you for your attention.

Sincerely yours,

**Reviewer 1:**

Overall, I am happy with the author's reply to my previous comments. Additionally, I suggest following corrections/revisions before publication.

**Response:**

*Thank you for your useful comments and hereby we present our responses for the queries below:*

1) The authors addressed the limitation of the study associated with 500-hPa top when aod is calculated. However, it is not stated in abstract and conclusion where it should be stated.

**Response:**

*Thank for your suggestion. We now add sentences to highlight this in abstract (lines 26-28, page 1) "AODEM (extended aerosol optical depth module) was used to calculate the total columnar aerosol optical depth (AOD) and BC AOD up to the top of the domain at 500 hPa (~5,500 m) which did not include the free tropospheric long-range transport of the pollution". Suggestion is added in the conclusion (lines 8-9, page 19) "Vertical model set-up should be extended beyond of 500 hPa (~5,500 m) in the future studies to better incorporate the free tropospheric LRT of aerosol".*

2) Page 7, lines 7-11: Rainfall patterns are shown, but similarity and/or difference between model and observation are never discussed. How does rainfall pattern affect simulated aod and/or pm?

**Response:**

*We actually have discussed the similarity and difference between the model and satellite observation in section 3.2.2 on "Synoptic scale model evaluation" page 10, lines 15-19 "The modeled monthly precipitations for two selected months (August and October, 2007) were compared with the TRMM-3B43 dataset in Figure 2 which showed good agreement in the distribution patterns but the model somehow underestimated the domain maximum monthly precipitation column, for example, that occurred over Myanmar in August 2007 or over the central part of Vietnam in October".*

*We now add explanation of the effect of precipitation on the simulated PM concentrations in section 3.4 lines 24-28, page 16 "Effects of precipitation on the PM levels were also seen, e.g. higher PM levels (Figure 7) were simulated over Indochina in January, October and November as compared to August because the latter was a rainy month in this part of the domain, i.e. less biomass open burning and more wet removal in principle. The opposite was actually seen in the Southern part of modeling domain, e.g. above Indonesia, where lower PM levels were simulated in October (more rainy month in this part) than other months".*

*Effect of precipitation on AOD is also added in lines 19-20, page 17* "Consistently with the PM results, the effects of precipitation on AOD were captured, i.e. higher in the dry months and lower in the wet months in the respective parts of the domain".

*Note that in this version we update the Figure 7 by adding results for the month of October 2007 and also Figure 8 by adding result of AOD in October 2007 for comparison with the monthly rainfall presented in Figure2.*

3) Page 10, 1st paragraph, & Fig. S4: Surface pressure fields from model are compared with observed Sea-level pressure. I suggest to use sea-level pressure for model as well.

**Response:**

*Thank you. In this version, we now compare the model results of surface pressure with the ERA interim reanalysis dataset and the figure S4 is revised accordingly. Discussion is added in lines 9-14, page 10* "Spatial distribution of surface pressure over the WRF domain is presented together with the ERA interim dataset in Figure S4 for three selected days (Jan 1, 2013; Oct 8, 2007; and Nov 7, 2007; 00:00 UTC). Both modeled and ERA data showed similar spatial distribution patterns of pressure but WRF appeared to produce slightly lower surface pressure over central Papua of Indonesia for all 3 cases presented. In fact, both datasets showed lower pressure zones over the high mountain areas of Himalaya, eastern parts of China and central Papua of Indonesia that indicated the effects of the topography".

4) Page 10, lines 11-17 & Fig. S5: Hard to compare two, but it seems that model and observation winds are quite different. Since WRF model has initial condition from NCEP reanalysis, can NCEP winds be used for observational counterpart. Also, "synoptic upper wind filed" is not commonly used terminology.

**Response:**

*Thank you for your suggestion. We now compare the upper wind (850 hPa) with the ERA interim reanalysis dataset and Figure S5 is revised accordingly and add new upper wind result for 1st January 2007. We change the "synoptic upper wind field" to "upper wind field" in Figure S5 caption.*

*Discussion is added in lines 18-21, page 10* "The simulated wind fields at 850 hPa (~ 1,500 m) are compared with the ERA interim upper wind fields in Figure S5 that also showed a consistency of the two datasets and more in the center of the domain both for wind speeds and wind directions. A large discrepancy was seen at the NW corner of the modeling domain and this may be attributed to the boundary conditions (taken from NCEP FNL in this study)".

**Reviewer 2:**

I would like to thank the authors for their overall revised content. I thought that they carefully answered the points from each reviewer. I would like to note some points that I am concerned about, including some points I did not notice in the previous review.

**Response:**

*Thank you for the useful comments/corrections from the reviewers that helped us to refine the manuscript. The responses to the queries given by reviewer are presented below:*

\<Abstract\> The authors commented that this paper was intended to be composed of Part 1 and Part 2 but would handle it as two separate papers. However, Permadi et al.'s title on the reference has not been modified, and it remains as Part 2. Although it is not directly related to this paper, I think that it is better to modify it.

**Response:**

*Thank you for your correction and we revised the references accordingly to "Permadi, D.A., Kim Oanh, N.T., Vautard, R.: Assessment of emission scenarios for 2030 and co-benefits of black carbon emission reduction measures on air quality and climate forcing in Southeast Asia, Submitted to Atmospheric Chemistry and Physics , 2017a.*

. \<text\> p.3 L7 "The results of this Paper 1 are used in the follow up study which investigated the potential co-benefits of various emission reduction measures implemented in Indonesia and Thailand on air quality improvement, number of premature death reduction and climate forcing mitigation in 2030 (Permadi et al., 2017a)." It remains a composition of Part 1 and Part 2.

**Response:**

*Sentence is revised in lines 31-34, page 1, abstract* " The results of this paper are used to calculate the regional aerosol direct radiative forcing under different emission reduction scenarios to explore potential co-benefits for air quality improvement, reduction in number of premature deaths and climate forcing mitigation in SEA in 2030 (Permadi et al., 2017a)".

p.3 16L There is a description of spatial distribution, but the description of OB is followed in this part. I think that it would be better to add a description on spatial distribution of OB or to put the description of spatial distribution after the OB description.

**Response:**

*Thank you. As suggested we moved the information of the spatial distribution after the parts explaining the OB emission in lines 4-13, page 4.*

p.3 L32 and p.4 L1 Explanation of GFED3 abbreviation is coming after the first appearance. An abbreviation should be added after the first case the term is used.

**Response:**

*Thank you for the correction and we moved the abbreviation of GFED3 to lines 21-22, page 3 when it appears for the first time.*

p.4 L5 The description of the ship is in the paragraph of CROB emission.

**Response:**

*Thank you and now we move this after the paragraph of biogenic emission in lines 1-2, page 4.*

p.4 L10 The listed URL (http: //glcf.umi aces.umd.edu) did not reach the GFCL site. Please check it.

**Response:**

*Thank you it is now updated to* http://glcf.umd.edu/ *in line 34. Page 3.*

p.4 L11 Is the expression "NOx emissions from natural vegetation." correct? "NO emissions from soil"?

**Response:**

*Thank you for your suggestion and we correct in line 1 page 4.*

p.6 L13 The authors compared boundary concentrations of 2007 and 1998-2002, and the difference was described as being in the range of 0.98 - 1.23. However, the authors' consideration about this value is not shown in the text. As in the "authors' response to reviewer", it needs a comment such as "This implies that basically the two datasets were almost similar. The impacts of the aged boundary conditions on the simulation are expected but with a small magnitude".

**Response:**

*Thank you and we add sentence in lines 16-18, page 6 as suggested "This implies that basically the two datasets were almost similar. The impacts of the aged boundary conditions on the simulation are expected but with a small magnitude".*

p.10 L14 Why do the authors show examples in October and November to compare the wind situation of the upper layer with observations?

**Response:**

*We would like to take to take snapshots of one period of dry season for the areas in the upper part of the equator line (in November) which includes Thailand and other countries located in the continental SEA and below the equator line (in October) which includes Indonesia and Timor Leste. We add now a selected day in Figure for the 1st January 2007, 07:00 LST for comparison to be consistent with the discussion on surface pressure and precipitation.*

p.10 L20 Please write the approximate altitude of 500 hPa as shown on p.10 L12.

**Response:**

*Thank you we add the approximation of the physical height of 500 hPa in (~5,500 m) line 22, page 10.*

p.13 L4 I think that it is better to explain this part "EC was measured using thermal optical method while BC was measured using light absorption method." at the measurement method (p.12 L29 or L30). In addition, please specify that there is no problem about comparison of EC and BC directly.

**Response:**

*Thank you. We move the explanation of EC and BC at the measurement method part in lines 32-34, page 12. We also add sentence to explain comparison between modeled BC and observed EC for AIT site in lines 4-5, page 13 "This is because for PM mass closure, EC seems to be better while BC is suitable for radiative transfer budget analysis (Gelencsér, 2004)".*

p.15 L10 Is the value shown in Table S3 the average value of the entire model domain? Please write it clearly.

**Response:**

*Thank you we revise the annual average in the Table S3 to be "Max annual avg" and add footnote "One maximum value simulated in the whole modeling domain for the considered period". The title of Table S3 is revised to "Summary of simulated domain maximum ground-level concentrations PM$_{10}$, PM$_{2.5}$ and BC for different periods".*

p.17 L13 The position of 10 AERONET stations can also be added in Figure S1. Or I think that it is good to put them in Figure 8.

**Response:**

*Thank you for your suggestion, we add in Figure S1 to be placed together with other ground-based observation.*

<Reference> p.24 L5 Is 4 of "UNEP-C4." a superscript?

**Response:**

*Thank you and you are correct it should be a superscript for The Center for Clouds, Chemistry and Climate ($C^4$). Corrected accordingly in the reference list "UNEP-$C^4$.: The Asian Brown Cloud: Climate and Other Environmental Impacts, UNEP, Nairobi, 2002".*

<Figures> Figure 1  The letter CO in b) has disappeared.

**Response:**

*Thank you for your correction. Figure 1 is revised accordingly.*

Figure 6 I am sorry I could not point it out in the last review, but I think that "PBCAR" in the legend should be "BC".

**Response:**

*We revised the legend of "PBCAR" to "BC" in Figure 6 accordingly. The word of PBCAR comes from the output of CHIMERE model.*

Figure 8

Legend numbers are too small to read.

**Response:**

*Thank you. Legend in Figure 8 is enlarged accordingly.*

Figure S4 The calculation result and the weather chart are difficult to compare. Since the calculation result is drawn with surface pressure, the information of the terrain in the figure is highlighted and it is difficult to compare the pressure pattern. Please draw to sea level pressure uniformly.

**Response:**

*Thank you. We now compare the modeled surface pressure with the ERA interim dataset and figure S4 is revised accordingly.*

Figure S7 As in Figure 7, it is better to write "PM 2.5 January" in the figure.

**Response:**

*Thank you. Figure S7 is revised accordingly. It is actually annual average of $PM_{2.5}$.*